# Transposable elements-mediated recruitment of KDM1A epigenetically silences HNF4A expression to promote hepatocellular carcinoma

Tiantian Jing[1,5], Dianhui Wei [1,5], Xiaoli Xu[1,5], Chengsi Wu[1], Lili Yuan[1], Yiwen Huang[1], Yizhen Liu [2] ✉, Yanyi Jiang [3,4] ✉ & Boshi Wang [1] ✉

Transposable elements (TEs) contribute to gene expression regulation by acting as cis-regulatory elements that attract transcription factors and epigenetic regulators. This research aims to explore the functional and clinical implications of transposable element-related molecular events in hepatocellular carcinoma, focusing on the mechanism through which liver-specific accessible TEs (liver-TEs) regulate adjacent gene expression. Our findings reveal that the expression of HNF4A is inversely regulated by proximate liver-TEs, which facilitates liver cancer cell proliferation. Mechanistically, liver-TEs are predominantly occupied by the histone demethylase, KDM1A. KDM1A negatively influences the methylation of histone H3 Lys4 (H3K4) of liver-TEs, resulting in the epigenetic silencing of HNF4A expression. The suppression of HNF4A mediated by KDM1A promotes liver cancer cell proliferation. In conclusion, this study uncovers a liver-TE/KDM1A/HNF4A regulatory axis that promotes liver cancer growth and highlights KDM1A as a promising therapeutic target. Our findings provide insight into the transposable element-related molecular mechanisms underlying liver cancer progression.

Transposable elements, or transposons, are mobile DNA units present in the majority of eukaryotic genomes. Although once considered "junk DNA" with no functional significance, recent studies have revealed that TEs provide regulatory sequences that wire transcriptional regulatory networks[1–3]. While host organisms usually silence TEs, the remnants of TE-derived cis-regulatory elements (CRE) can persist and adapt to control the transcription of host genes[4–7].

TEs interact with cancer in intricate ways, offering novel insights into the mechanisms of cancer development and progression[8,9]. TEs can function as promoters, driving oncogene expression and oncogenesis, which contribute to tumor initiation and maintenance[10]. They can establish regulatory circuits involving the KRAB zinc-finger protein family to suppress tumors[11]. Moreover, TEs can interplay with epigenetic events such as DNA methylation and histone modifications to modulate gene expression patterns and promote tumor growth[12]. The involvement of TEs in tumoral epigenetic regulation represents a consequential outcome of TE silencing to maintain genome stability. A primary mechanism for suppressing TEs involves the selective deposition of repressive histone modifications[13]. Tumor cells employ this strategy to facilitate immune evasion[14,15]. Additional epigenetic

[1]State Key Laboratory of Systems Medicine for Cancer, Shanghai Cancer Institute, Renji Hospital, Shanghai Jiao Tong University School of Medicine, Shanghai 200032, China. [2]Department of Medical Oncology, Fudan University Shanghai Cancer Center; Department of Oncology, Shanghai Medical College, Fudan University, Shanghai 200032, China. [3]Institute of Health and Medical Technology, Hefei Institutes of Physical Science, Chinese Academy of Sciences, Hefei 230031, China. [4]University of Science and Technology of China, Hefei 230026, China. [5]These authors contributed equally: Tiantian Jing, Dianhui Wei, Xiaoli Xu. ✉e-mail: aliuyz@126.com; yanyij@cmpt.ac.cn; wbs137@shsci.org

regulators, like the human silencing hub (HUSH) complex and lysine demethylase 1A (KDM1A), employ a similar TE-dependent mechanism to impact tumor development[16–18].

KDM1A, a well-established oncogene, exhibits high expression in multiple cancers and correlates with unfavorable prognosis in cancer patients[19,20]. In liver cancer, KDM1A promotes cancer growth and drug resistance by regulating signals such as FKBP8/Bcl2 or LINC01134/SP1/p62[21–24]. KDM1A typically functions as part of a complex, such as the CoREST complex, where it selectively removes H3K4me1 and H3K4me2 methylation modifications to inhibit transcription of target genes[25,26]. Alternatively, in a complex with hormone receptors, KDM1A specifically regulates H3K9me2 and H3K9me3 methylation modifications[27,28]. The selectivity of KDM1A towards histone substrates depends on the composition of the complex[20,28–30]. HNF4A, a member of the nuclear receptor superfamily, serves as a pivotal regulatory factor in the initial stages of liver cancer development[31–33]. In normal physiological conditions, HNF4A exhibits high expression in liver tissue and plays a critical role in preserving the differentiation and function of liver cells[31,34]. However, HNF4A expression is downregulated in liver cancer and shows tumor-suppressive characteristics[35].

Here, we show a mechanism in which KDM1A epigenetically suppresses transposable elements in proximity to the *HNF4A* gene, resulting in the inhibition of HNF4A expression. This work focuses on liver-specific accessible TEs (liver-TEs) that provide CRE and recruit KDM1A to silence nearby genes. Moreover, we discover a signaling axis in which liver-TEs, KDM1A, and the liver-TE-associated gene (*HNF4A*) synergistically control the proliferation of hepatocellular carcinoma cells. These findings elucidate the functional and clinical implications of transposable element-mediated molecular events, potentially paving the way for novel therapeutic strategies in hepatocellular carcinoma.

## Results

### Liver-TEs function as inhibitory cis-regulatory elements for the HNF4A gene

To explore the involvement of transposable elements (TEs) as cis-regulatory elements in liver cancer development, we utilized the TCGA ATAC-seq data from 23 different types of cancer tissues to identify specific accessible TEs within transcriptional regulatory regions (TRR: TSS ± 10 kb) in liver cancers. Our analysis unveiled significant heterogeneity in TE accessibility across distinct tumor types. Through ATAC-seq data analysis, we identified a total of 3,762 TEs exhibiting elevated ATAC-seq intensity within liver cancer samples (Fig. 1a, b and Supplementary Data 1). By using ATAC-seq data from non-malignant human tissue samples in the ENCODE database, we found that the identified TEs were highly specific to healthy liver tissues as well (Fig. 1c). Therefore, we designated this subset of TEs as liver-specific accessible TEs (liver-TEs). Concurrently, the transcriptional regulatory regions harboring liver-TEs (designated as liver-TE-TRR) in liver cancer samples exhibited a noticeable increase in ATAC-seq signal intensity (Supplementary Fig. 1a and Supplementary Data 1). To further elucidate the distinctions in the accessibility of liver-TEs and liver-TE-TRRs between healthy and cancerous liver cells, we conducted ATAC-seq assays in both cell types. The results revealed that liver-TEs and liver-TE-TRRs were more accessible in normal liver cells (Supplementary Fig. 1b).

Liver-TEs comprised major TE families, including SINEs (Alu, MIR), LINEs (L1, L2), LTRs (ERV1, ERVL, ERVL-MaLR), and DNA (hAT-Charlie, TcMar-Tigger). Upon comparing the proportions of significant TE families within the human genome and liver-TEs, we observed the highest representation of the Alu family and MIRs (Mammalian-wide interspersed repeats) in liver-TEs, with a notably augmented proportion of MIR family constituents (Supplementary Fig. 1c). Genes located within ±10 kb of liver-TEs were defined as liver-TE associated genes

(Supplementary Data 2). Analysis of gene expression profiles using the combined GETx-TCGA database revealed that liver-TE-associated genes exhibited liver tissue specificity, both healthy and cancerous, with a slight downregulation in liver cancer compared to normal liver tissues (Supplementary Fig. 1d). Furthermore, prognosis analysis of liver-TE-associated gene expression demonstrated that high expression of these genes indicated a favorable prognosis for liver cancer patients (Fig. 1d and Supplementary Fig. 1e). Gene function enrichment analysis of liver-TE-related genes indicated that they are associated with liver-specific gene expression, and are enriched for HNF4A signature (Fig. 1e). Intriguingly, the P1-driven *HNF4A*, a predominant isoform within liver tissues[36], exhibits two liver-TE clusters flanking the transcription start site (TSS) (Fig. 1f). To explore the regulatory impact of liver-TEs on HNF4A expression, we utilized CRISPR/Cas9 technology to concurrently remove each cluster of TEs, situated either upstream or downstream of the P1-driven *HNF4A* TSS, referred to as HNF4A-liver-TEs, in liver cancer cells (Supplementary Fig. 1f). Based on ATAC-seq data, the majority of HNF4A-liver-TEs exhibited enhanced accessibility in normal liver cells (Supplementary Fig. 1g). Through the analysis of Capture Hi-C data, we identified that genomic interactions involving HNF4A-liver-TEs predominantly involve cis-interactions, with a notable concentration in the q13.12 region of chromosome 20 (Supplementary Fig. 1h). Deletion of liver-TEs resulted in an upregulation of HNF4A expression (Fig. 1f-g), thus illustrating the inhibitory role of TE elements in controlling HNF4A expression. To gain deeper insights into the regulatory mechanisms, we employed gene editing techniques to individually excise several long HNF4A-liver-TEs. This approach enabled us to precisely elucidate the specific roles of liver-TEs on the *HNF4A* gene, revealing that depletion of a single TE element plays a role in regulating HNF4A expression (Supplementary Fig. 1i). Furthermore, liver cancer cells deficient in HNF4A-liver-TEs exhibited diminished growth capacity both in vitro (Fig. 1h) and in vivo (Supplementary Fig. 1j). Similarly, the depletion of HNF4A-liver-TEs in HCC patient-derived organoids resulted in growth retardation (Supplementary Fig. 1k). These observations underscore the significant role of this element in promoting tumorigenesis.

### KDM1A is enriched in liver-TEs and is necessary for HCC cell growth

To investigate the epigenetic characteristics of liver-TEs, we analyzed the histone marks enriched in liver-TEs using the RemapEnrich algorithm. The results showed that liver-TEs are enriched in active histone marks such as H3K4me1, H3K4me2, and H3K27ac (Supplementary Fig. 2a, Supplementary Data 3). The enrichment of H3K4 methylation marks in liver-TE regions suggests their presence in relatively accessible chromatin environments. Additionally, our investigation of CpG methylation levels in liver-TEs showed that they were comparable to those in non-liver-TEs (Supplementary Fig. 2b), suggesting that DNA methylation does not significantly influence liver-TE activity.

To elucidate the molecular mechanisms underlying the regulation of gene expression by liver-TE, we performed a RemapEnrich analysis of transcriptional regulators (TR) enrichment on liver-TEs. Our results reveal a substantial enrichment of the histone demethylase KDM1A on liver-TEs (Fig. 2a and Supplementary Data 4). Heatmaps displaying the KDM1A enrichment signals within liver-TEs demonstrate a significant increase in KDM1A signals within these regions compared to similarly sized shuffled TE regions (Fig. 2b). Then, we conducted KDM1A CUT&Tag-seq assays in both normal liver cells (THLE2, THLE3) and HCC cell lines to compare the binding ability of KDM1A within liver-specific TEs. Our results demonstrated a diminished binding capability of KDM1A to these TEs in normal liver cells compared to HCC cells (Supplementary Fig. 2c), suggesting potential aberrant recruitment of KDM1A to liver-specific TEs within liver cancer cells. To test whether KDM1A plays an epigenetic regulatory role within liver-TEs, ATAC-seq and CUT&Tag-seq experiments were conducted. The results revealed

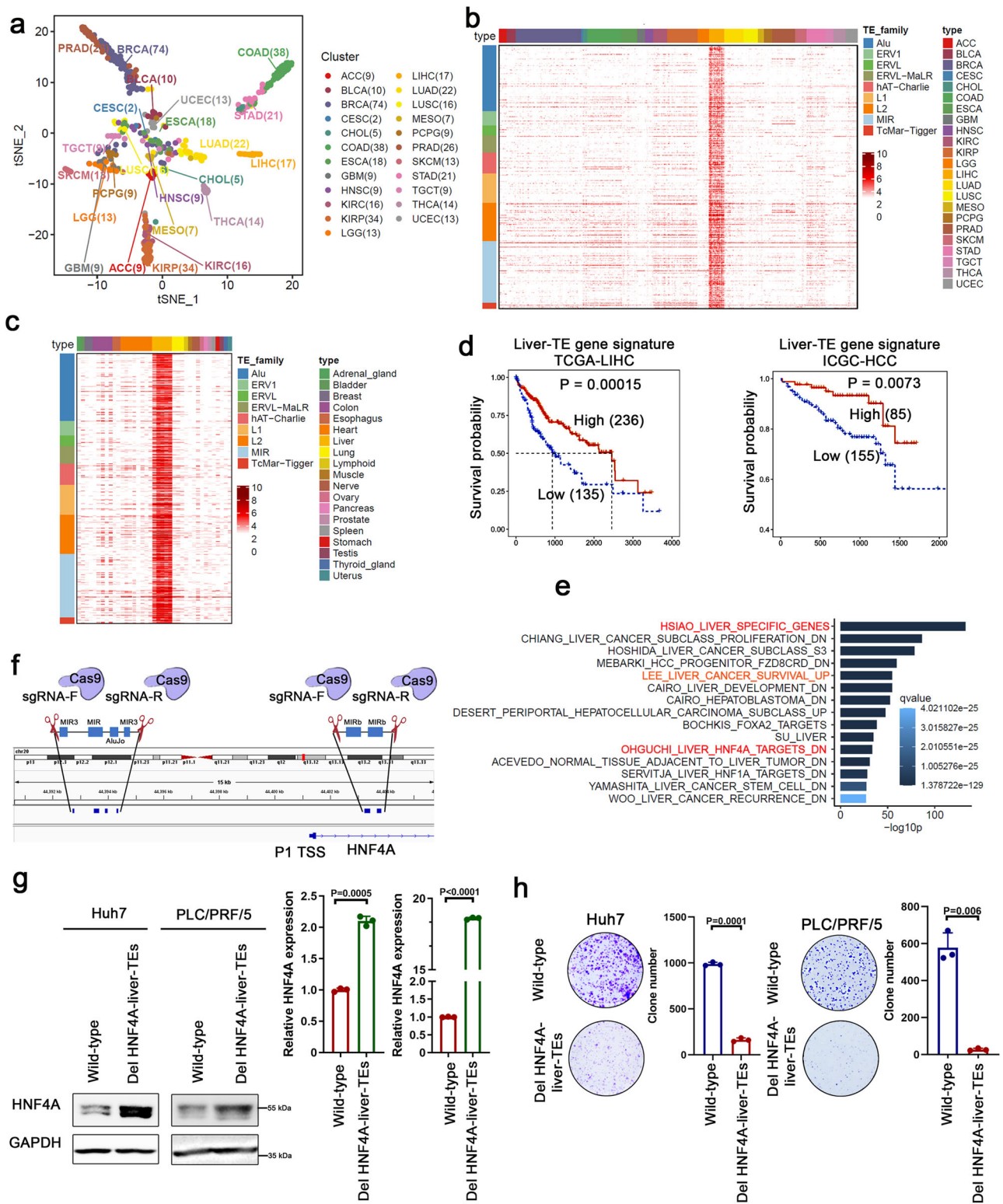

that KDM1A knockdown led to an increase in chromatin accessibility (Fig. 2c) and histone mark H3K4me1, H3K4me2, H3K9me2, and H3K27ac within liver-TEs (Fig. 2d). However, the global histone methylation modifications remained largely unaffected by KDM1A knockdown (Supplementary Fig. 2d). These results indicate that KDM1A downregulation enhanced the activation and accessibility of liver-TEs and nearby chromatin. Additionally, we conducted a detailed examination of how KDM1A knockdown influences H3K4me1 and H3K9me2 marks in the transcriptional regulatory regions bound by

KDM1A. Our results demonstrate that in regions directly targeted by KDM1A, H3K4me1 exhibited upregulation following KDM1A knockdown, while H3K9me2 showed no significant change (Supplementary Fig. 2e, Top panel). This observation is consistently supported by findings in liver-TE-associated KDM1A-targeted TRRs, where H3K4me1 marks also significantly increased upon KDM1A knockdown (Supplementary Fig. 2e, Down panel). Notably, the reduction of KDM1A in normal liver cells did not lead to an elevation in H3K4me1 modification of liver-TEs, which lack KDM1A binding peaks (Supplementary Fig. 2f).

**Fig. 1 | Negative regulation of HNF4A expression by liver-TEs. a** tSNE dimensional reduction analysis demonstrates heterogeneity in TE accessibility across different tumor types. **b**, **c** The heatmap displays the log-transformed ATAC-seq signal intensity of the TEs highly accessible in liver cancer samples (**b**) and non-malignant liver tissues (**c**). **d** The expression (GSVA score) of liver-TE-associated genes in TCGA-LIHC and ICGC-HCC liver cancer transcriptome datasets is correlated with a favorable prognosis for patients. The survival rates were compared using a two-sided log-rank test, without adjusting for multiple comparisons. n = 371 samples in TCGA-LIHC dataset, High = 236, Low = 135. n = 240 samples in ICGC-HCC dataset, High = 85, Low = 155. **e** Gene function enrichment analysis of liver-TE-associated genes was performed using the MSigDB gene set database and the R package Clusterprofiler. The one-sided hypergeometric test was used to assess

gene set enrichment and *p*-values were adjusted using the Benjamini–Hochberg (BH) method. The bar plot displays the top 15 significantly enriched functional gene sets. **f** Schematic diagram shows the CRISPR/Cas9-mediated depletion of HNF4A-liver-TEs within HNF4A TRR. **g** HNF4A expression was determined by real-time PCR and Western blot assays after CRISPR/Cas9-mediated depletion of HNF4A-liver-TEs in liver cancer cells, n = 3 biological replicates. Significance was examined by a two-sided t-test, and mean ± SD was shown. The experiments were repeated three times with consistent results. **h** Colony formation assay demonstrated the growth-inhibitory effect of depleting HNF4A-liver-TEs, n = 3 biological replicates. Significance was examined by a two-sided t-test, and mean ± SD was shown. Source data are provided as a Source Data file.

These results strongly suggest that KDM1A may directly regulate H3K4me1, leading us to focus our subsequent analyses on this mark.

To further confirm our findings, we treated HepG2 cells with several KDM1A small-molecule inhibitors and observed similar changes in H3K4me1 as seen in KDM1A knockdown cells (Supplementary Fig. 2g). We found that out of a total of 1923 liver-TE associated genes, 1202 genes were targeted by KDM1A, accounting for approximately 62.5% of the cohort (Fig. 2e). RNA-seq and GSEA assays demonstrated that knockdown of KDM1A resulted in a global elevated expression of liver-TE-associated genes (Fig. 2f), suggesting a global inhibitory role of KDM1A in the regulation of this set of genes and coordinating the inhibitory role of KDM1A toward liver-TEs.

Colony formation assays demonstrated significant inhibition of liver cancer cell growth upon KDM1A downregulation (Fig. 2g), while KDM1A overexpression promoted cell growth (Supplementary Fig. 2h). To assess the relative importance of KDM1A in liver cancer cell growth compared to other histone methylation regulators, we ranked their mean dependency scores derived from RNAi screen results in the DepMap database. The results indicated a high dependency of liver cancer cells on KDM1A (Supplementary Fig. 2i and Supplementary Data 5). Furthermore, pharmacological inhibiting KDM1A with small-molecule inhibitors dramatically attenuating liver cancer cell growth (Supplementary Fig. 2j). Subcutaneous tumorigenicity assays in nude mice demonstrated that KDM1A knockdown significantly inhibited the in vivo tumorigenicity of liver cancer cells (Supplementary Fig. 2k). Strikingly, we conducted experiments using HCC-PDX model to assess the in vivo inhibitory effects of the KDM1A inhibitor SP2509. Our results demonstrated significant retardation of tumor growth upon SP2509 treatment, while no alterations in body weights were observed (Fig. 2h). Subsequently, we analyzed tumor samples from these experiments and observed an increase in the H3K4me1 within liver-TEs following KDM1A inhibition (Fig. 2i). Concurrently, we cultured PDX samples in vitro to establish a patient-derived cell line. Depletion of HNF4A-liver-TEs and KDM1A knockdown exerted notable suppressive effects on tumor cell growth (Supplementary Fig. 2l), consistent with the observed effects in conventional HCC cell lines. Moreover, by utilizing the HCC-PDO model, we investigated the impact of HNF4A-liver-TEs depletion and KDM1A inhibition on the 3D growth of HCC cells, revealing significant inhibition of 3D growth upon KDM1A inhibition (Fig. 2j). Taken together, the above findings indicate that targeting KDM1A could be a promising approach to regulate liver-TE activity and a potential strategy for the treatment of HCC.

## KDM1A regulates HNF4A expression via liver-TEs

We demonstrated that *HNF4A* is a liver cancer suppressor gene negatively regulated by liver-TEs. This leads us to hypothesize that KDM1A can regulate the expression of the *HNF4A* gene through liver-TEs. This hypothesis was substantiated through subsequent experiments. Real-time PCR and Western blot experiments validated the regulatory role of KDM1A in HNF4A expression, as KDM1A downregulation elevated mRNA and protein levels of HNF4A (Fig. 3a and Supplementary Fig. 3a). Similarly, the application of SP2509, a KDM1A inhibitor, heightened

HNF4A expression in liver cancer cells (Fig. 3b). Conversely, the overexpression of KDM1A led to a reduction in HNF4A expression (Fig. 3c and Supplementary Fig. 3b). The HCC-PDX samples treated with SP2509 also exhibited elevated expression levels of HNF4A (Supplementary Fig. 3c). Additionally, gene function enrichment analysis revealed that KDM1A negatively regulated the downstream gene signature of HNF4A (Fig. 3d). Further examination of histone marks around the *HNF4A* gene revealed that KDM1A knockdown enhanced H3K4me1 modification levels (Fig. 3e and Supplementary Fig. 3d). Given that H3K4me1 is a direct demethylation substrate of KDM1A, our results suggest that KDM1A inhibits HNF4A expression by promoting H3K4me1 demethylation. To ascertain whether KDM1A-mediated regulation of HNF4A expression relies on its catalytic activity, we reintroduced either wild-type or enzymatically inactive K661A mutant KDM1A into KDM1A knockdown cells. The results indicated that while wild-type KDM1A effectively counteracted the elevated HNF4A expression induced by KDM1A knockdown, the K661A mutant failed to produce a similar effect (Supplementary Fig. 3e). Additionally, in addition to reversible KDM1A inhibitors, irreversible inhibitors that selectively block the catalytic activity of KDM1A also increased HNF4A expression (Supplementary Fig. 3f). These findings suggest that the catalytic activity of KDM1A is pivotal for its regulation of HNF4A expression.

To explore whether HNF4A-liver-TEs is essential for KDM1A-mediated inhibition of HNF4A expression, we overexpressed KDM1A in HCC cells depleted of HNF4A-liver-TEs, which resulted in the loss of the ability of KDM1A to suppress HNF4A expression (Fig. 3f). Correspondingly, the binding affinity of KDM1A to the transcriptional regulatory region of the *HNF4A* gene was weakened upon the removal of HNF4A-liver-TEs (Fig. 3g). Furthermore, the tumor-promoting effects of KDM1A were also attenuated by HNF4A-liver-TE deletion (Fig. 3h).

In consideration of the pivotal role played by HNF4A-liver-TEs in mediating the regulatory functions of KDM1A, it was postulated that exposure to HNF4A-liver-TEs may facilitate the recruitment of KDM1A. To interrogate this hypothesis, we employed CRISPRa technology to manipulate HNF4A-liver-TE, observing a consequent increase in the binding affinity of KDM1A (Supplementary Fig. 3g), concomitant with the repression of HNF4A expression (Supplementary Fig. 3h). Subsequent assessment of chromatin state revealed a reduction in chromatin accessibility, alongside decreases in H3K27ac and H3K4me1 modifications, at both HNF4A-liver-TEs and the HNF4A transcription start site (Supplementary Fig. 3i). These findings suggest that despite the action initiated by CRISPRa, the subsequent inhibitory effects of recruited KDM1A prevail. This is evidenced by observations indicating that knockdown of KDM1A in cells with CRISPRa targeting HNF4A-liver-TEs counteracted the loss of chromatin accessibility and reductions in H3K27ac and H3K4me1 (Supplementary Fig. 3i), ultimately alleviating the downregulation of HNF4A expression (Supplementary Fig. 3j). These collective observations highlight the predominant role of KDM1A in mediating the transcriptional inhibition of HNF4A induced by HNF4A-liver-TEs.

In non-malignant liver cells, where KDM1A exhibited lower binding efficiency to liver-TEs (Supplementary Fig. 2c), neither disruption

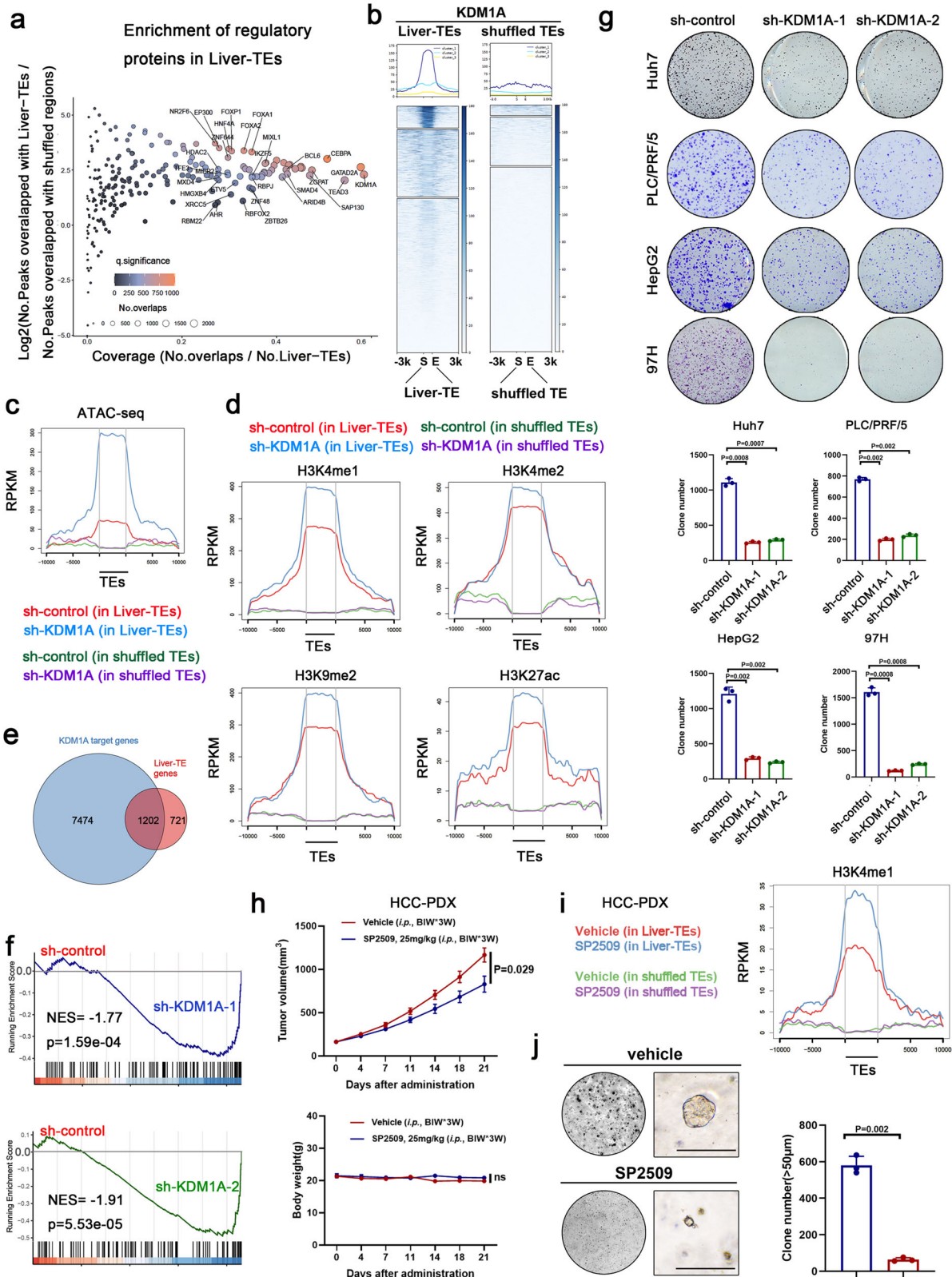

of liver-HNF4A-TEs nor inhibition of KDM1A influenced HNF4A expression (Supplementary Fig. 3k, l). To ascertain whether the promoting effect of KDM1A on liver cancer depends on its regulation of HNF4A, we simultaneously silenced KDM1A and HNF4A in liver cancer cells. We found that the restrained in vitro and in vivo cell growth caused by KDM1A knockdown was partially restored by HNF4A downregulation (Supplementary Fig. 4a, b), indicating that KDM1A

promotes the growth of liver cancer cells by inhibiting the expression of HNF4A.

## KDM1A interacts with the HNF4A complex to epigenetically silence its downstream genes

Analysis of ChIP-seq experimental data revealed a significant positive correlation in the binding intensities of TRR between KDM1A, HNF4A,

**Fig. 2 | Inhibition of KDM1A regulates liver-TEs and impedes HCC cell growth.**
**a** Analysis of the enrichment level of transcription regulatory (TR) proteins on liver-TE elements using ReMapEnrich software, which employed a two-sided binomial test, with $p$-values adjusted for multiple comparisons using the Benjamini–Yekutieli (BY) method. x: The number of overlaps/total number of liver-TEs, y: Log2 (No. Peaks overlapped with liver-TEs/No. Peaks overlapped with shuffled regions). The exact significance value for each TR was provided in Supplementary Data 4.
**b** Heatmap showing the ChIP-seq profiles of KDM1A within the liver-TE ± 3 kb regions. The control is a set of randomly shuffled regions of the same length as the liver-TEs. **c** Ngsplot shows the ATAC-seq signal intensity within liver-TEs and the ±10 kb regions in HepG2 cells with downregulated KDM1A. The signal intensity within randomly shuffled TEs was used as control. **d** CUT&Tag-seq experiments for H3K4me1, H3K4me2, H3K9me2, and H3K27ac were performed in KDM1A down-regulated HepG2 cells. Ngsplot was utilized to illustrate changes in histone modifications in liver-TEs and the ±10 kb regions. The signal intensity within randomly shuffled TEs was used as control. **e** Venn plot shows the liver-TE-associated genes targeted by KDM1A. **f** RNA-seq experiment and GSEA analysis show the effect of knocking down KDM1A on the expression of a single set of liver-TE-associated

genes. The two-sided GSEA analysis was performed without $p$-value adjustment. **g** Colony formation assay was performed to assess the growth-inhibitory effects of KDM1A knockdown on liver cancer cells, n = 3 biological replicates. Statistical analyses were performed using a two-sided t-test, and mean ± SD was shown. **h, i** Nude mice harboring HCC-PDX (approximately 163 mm³) were randomly divided into 2 groups: one group received intraperitoneal injection of 10 μl/g vehicle (10% DMSO, 90% corn oil), and another group received SP2509 treatment (i.p., 10 μl/g, 25 mg/kg, twice a week for 3 weeks). Subsequently, tumor xenograft volumes and mice body weights were measured every three days for three weeks. Statistical analyses were performed using two-sided ANOVA, n = 6 biological replicates, and mean ± SEM was shown (**h**). After the mice were euthanized, HCC-PDX tumor tissues were tested by CUT&Tag-seq assays using H3K4me1 antibody, showing an increase of H3K4me1 modification within liver-TEs upon SP2509 treatment (**i**). **j** SP2509 treatment reduced the organoid formation ability of primary HCC cells, n = 3 biological replicates. Significance was examined by a two-sided t-test, and mean ± SD was shown. Bar = 100 μm. Source data are provided as a Source Data file.

and other transcription factors (HNF1A, FOXA3, and GATA4), which play regulatory roles in liver development (Supplementary Fig. 4c–e). These observations suggest a potential interaction between KDM1A and the HNF4A transcription complex. To test this hypothesis, we conducted Co-IP experiments and confirmed that endogenous KDM1A interacts with HNF4A in liver cancer cells (Fig. 4a). Moreover, we observed an interaction between exogenously expressed KDM1A and HNF4A in 293T cells (Fig. 4b). We also identified interactions between HNF4A and other entoderm transcription factors (Supplementary Fig. 4f). Additionally, these transcription factors, including HNF4A, HNF1A, FOXA3, and GATA4, were found to interact with KDM1A (Fig. 4c). Immunofluorescence experiments using laser confocal imaging demonstrated colocalization of KDM1A and HNF4A within the nuclei (Fig. 4d). Further analysis of RNA-seq data in hepatocellular carcinoma with KDM1A downregulation revealed significant enrichment of conserved motifs of HNF4A in the promoters of KDM1A-negative regulatory genes (HOMER: P = 1e−14, Rank 1) (Fig. 4e and Supplementary Data 6). Based on the ChIP-seq results, the binding peaks of KDM1A and other known HNF4A partners were enriched around HNF4A motifs (Fig. 4f). These findings indicate that KDM1A interacts with HNF4A to suppress the expression of HNF4A target genes.

To investigate the epigenetic regulation of KDM1A on HNF4A target genes, we performed CUT&Tag-seq experiments for H3K4me1, H3K4me2, H3K9me2, and H3K27ac in HepG2 cells with KDM1A knockdown or inhibition. We observed that the intensities of H3K4me1 surrounding HNF4A motifs and binding sites were upregulated upon KDM1A knockdown or inhibition (Fig. 4g, h). Furthermore, KDM1A downregulation or inhibition was accompanied by enhanced sequencing signals of H3K27ac-CUT&Tag in the aforementioned regions (Fig. 4g, h), indicating that KDM1A repressed the transcription of HNF4A target genes by removing H3K4me1 within TRR and altering chromosomal accessibility.

## KDM1A inhibits the expression of the HNF4A target gene
Real-time PCR analysis of several known downstream genes of HNF4A (Fig. 5a, b) and luciferase reporter assays (Fig. 5c) demonstrate that KDM1A knockdown or pharmacological inhibition activates the transcriptional regulatory activity of HNF4A. As anticipated, the downregulation of HNF4A in KDM1A knockdown cells counteracted the elevation of HNF4A target genes induced by KDM1A knockdown (Supplementary Fig. 5a, b). To identify liver cancer suppressor genes related to liver-TE that are controlled by KDM1A-mediated HNF4A repression, we focused on MAT1A, a known HNF4A target gene associated with liver-TE which is directly repressed by KDM1A (Fig. 5d and Supplementary Fig. 5c). Pan-cancer expression profiling showed that

MAT1A is liver tissue-specific, highly expressed in normal liver and liver cancer tissues, but downregulated in liver cancers compared to normal liver tissues (Supplementary Fig. 5d), a pattern exemplifies HNF4A downstream signature (Supplementary Fig. 5e). MAT1A is a key rate-limiting enzyme in methionine metabolic pathways and functions as a liver cancer suppressor[37]. Observation by the IGV browser revealed that the TRR of *MAT1A* is bound by KDM1A, HNF4A, HNF1A, FOXA3, and GATA4, wherein the H3K4me1 intensities were upregulated upon KDM1A knockdown, accompanied by an upregulation of *MAT1A* mRNA expression levels revealed by RNA-seq assays (Fig. 5e). Real-time PCR and Western blot analysis further confirmed that inhibition of KDM1A effectively enhanced MAT1A expression (Fig. 5f, g). Mass spectrometry analysis displayed a decrease in the upstream metabolites of methionine and sulfur amino acids (MSO and MET), which are catalyzed by MAT1A, and an increase in the downstream products SAM and SAH upon KDM1A inhibition (Fig. 5h), confirming the negative regulation of MAT1A by KDM1A. This suggests that KDM1A downregulates the activity of the HNF4A-regulated hepatocyte-specific methionine metabolic pathway, ultimately promoting liver cancer.

## ZMYM3 mediates the binding of liver-TE by KDM1A
We hypothesized that KDM1A requires an adapter protein with specific DNA-binding activity to recognize liver-TEs, as KDM1A lacks a DNA-binding domain. By using an IP-LC-MS approach, we identified eight DNA-binding proteins that are present in KDM1A immunoprecipitants in liver cancer cells (Fig. 6a). To determine which factor has the potential to assist KDM1A for DNA binding, we integrated RNA-seq results from KDM1A knockdown cells, public ChIP-seq results and the Lisa algorithm. The results showed that ZMYM3 was enriched in the gene transcription regulatory regions that were directly negatively regulated by KDM1A (the gene whose TRR region possessed KDM1A binding peaks and which was upregulated upon KDM1A deletion based on RNA-seq assays) (Fig. 6b). ZMYM3 is a zinc-finger protein with a DNA-binding domain, and its interaction with KDM1A may contribute to the recruitment of KDM1A to the target DNA regions. The interaction was confirmed by endogenous Co-IP experiments (Fig. 6c). Additionally, similar to the in vivo impact of KDM1A, ZMYM3 knockdown also impeded tumorigenesis (Supplementary Fig. 6a). Peak intensity analysis showed that ZMYM3, KDM1A and HNF4A were positively correlated in the TRR (Fig. 6d and Supplementary Fig. 6b), indicating their co-occupancy in the genome. Moreover, ZMYM3, KDM1A, and HNF4A exhibited stronger binding ability to TRRs containing liver-TE compared to those without liver-TEs (Fig. 6d and Supplementary Fig. 6c).

To investigate whether ZMYM3 mediates KDM1A-induced demethylation of liver-TE, we performed CUT&Tag-seq experiments on KDM1A-overexpressing cells with or without ZMYM3 knockdown

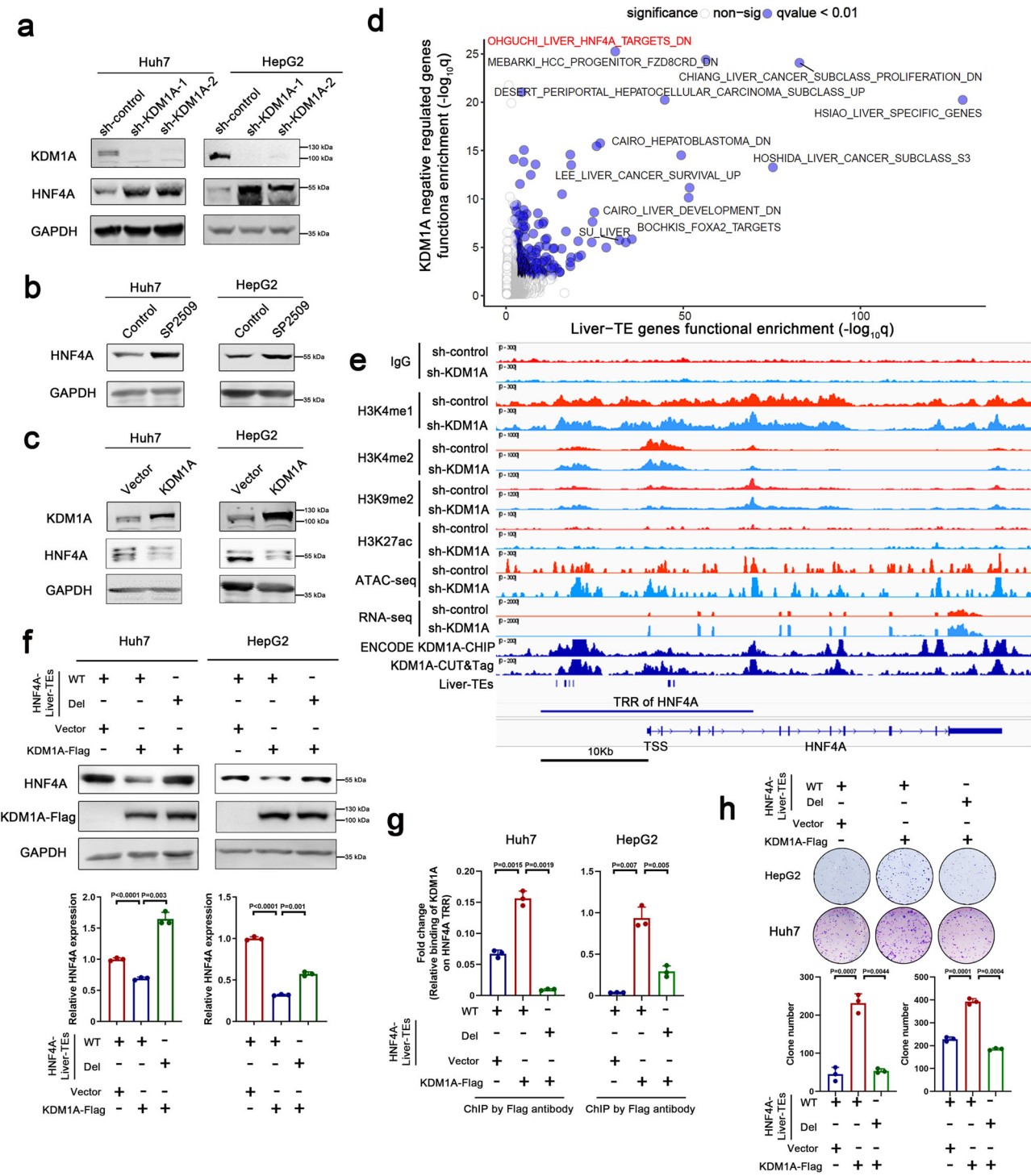

(Vector+sh-control, KDM1A+sh-control and KDM1A+sh-ZMYM3, respectively, Supplementary Fig. 6d). We observed the alterations in the KDM1A binding intensity and H3K4me1 modification both at KDM1A/ZMYM3 binding sites (Fig. 6e) and at liver-TEs (Fig. 6f). The results showed that ZMYM3 is required for the DNA-binding ability of KDM1A, as well as its specific binding to liver-TEs, as the binding of KDM1A to these regions decreased after ZMYM3 knockdown. Consistently, the H3K4me1 modifications on KDM1A/ZMYM3 binding regions and liver-TEs decreased after KDM1A overexpression, which was recovered by ZMYM3 knockdown. As ZMYM3 binding peaks are also correlated with HNF4A, we tested the roles of ZMYM3 in KDM1A-mediated demethylation of HNF4A binding regions. The results

showed that knockdown of ZMYM3 reversed the effects of KDM1A overexpression on HNF4A binding sites (Supplementary Fig. 6e). Functionally, knocking down ZMYM3 inhibits KDM1A-mediated growth of liver cancer cells (Fig. 6g). These results suggested that ZMYM3 is required for the epigenetic regulatory role of KDM1A and is necessary for KDM1A to exert its pro-growth effect in liver cancer cells.

To investigate the mechanism underpinnings of ZMYM3-mediated facilitation of KDM1A DNA binding, particularly within liver transposable elements (TEs), we posited that these liver-TEs might harbor or be in close proximity to specific DNA sequences recognized by the ZMYM3 protein. To evaluate this hypothesis, we utilized the MEME-ChIP program to identify ZMYM3 motifs and mapped their locations across the

**Fig. 3 | Epigenetic inhibition of HNF4A expression by KDM1A. a–c** Western blot analysis was conducted to assess the expression of HNF4A following KDM1A knockdown (**a**), SP2509 treatment (**b**), or KDM1A overexpression (**c**). The experiments were repeated three times, yielding consistent results. **d** Scatterplot displays results of gene functional enrichment analysis for liver-TE-related genes (n = 1923 genes) and KDM1A negatively regulated genes (n = 2346 genes). The one-sided hypergeometric test was used to assess gene set enrichment and p-values were adjusted using the Benjamini–Hochberg (BH) method. The horizontal and vertical axes represent the significance of the two groups of functional enrichment analysis (−log(q-value)). The KDM1A negatively regulated genes in RNA-seq results were defined as the genes significantly upregulated in both sh-KDM1A-1 and sh-KDM1A-2 (Foldchange >2, P < 0.05) expressed HepG2 cells. **e** CUT&Tag-seq for H3K4me1, H3K4me2, H3K9me2, H3K27ac, and ATAC-seq assays were performed in HepG2 cells with or without KDM1A knockdown, n = 1 per assay per condition. The status of these histone modifications in the *HNF4A* gene transcriptional regulatory region

was shown in the IGV genome browser. The IGV browser also displayed the RNA-seq data and KDM1A binding peaks (CUT&Tag-seq and ChIP-seq) in the *HNF4A* gene window. **f** Huh7 and HepG2 cells with HNF4A-liver-TEs deletion were overexpressed with KDM1A, and the expression of HNF4A was evaluated using Western blotting and real-time PCR, n = 3 biological replicates. Significance was examined by a two-sided t-test, and mean ± SD was shown. The experiments were repeated three times with similar results. **g** ChIP-PCR assays were performed in KDM1A-Flag expression Huh7 and HepG2 cells using Flag antibody to detect the binding of KDM1A near HNF4A-liver-TEs, n = 3 biological replicates. Significance was examined by a two-sided t-test, and mean ± SD was shown. **h** Colony formation assays were performed in Huh7 and HepG2 cells to assess the impact of HNF4A-liver-TEs deletion on KDM1A-mediated cell growth, n = 3 biological replicates. Significance was examined by a two-sided t-test, and mean ± SD was shown. Source data are provided as a Source Data file.

human genome. We then computed the distances between liver-TEs and the identified ZMYM3 motifs. This analysis revealed a notable overlap or close proximity between the majority of ZMYM3 motifs and liver-TEs (Supplementary Fig. 6f). Importantly, the distances between liver-TEs and ZMYM3 motifs were statistically closer than those observed between a randomly shuffled set of TEs and the motifs (Wilcoxon test, P < 2.2e−16). Subsequently, we provided additional evidence by assessing the KDM1A binding capacity in regions with or without ZMYM3 motifs within or near KDM1A peaks. Our findings revealed a significantly heightened binding capacity of KDM1A in regions containing ZMYM3 motifs (Supplementary Fig. 6g), suggesting a potential role of ZMYM3 in enhancing the DNA-binding capability of KDM1A.

### High expression of KDM1A is associated with poor prognosis in hepatocellular carcinoma

We performed immunohistochemical staining and survival analysis of KDM1A using 90 hepatocellular carcinoma tissue samples and found that high expression of KDM1A indicated poor prognosis (Fig. 7a, b). Furthermore, survival analysis using TCGA, ICGC, and GSE14520 datasets showed that the overall survival (OS) and progression-free survival (PFS) of patients in the high KDM1A expression group were significantly shorter than those in the low expression group (Fig. 7c). To corroborate our molecular findings with clinical samples, we conducted immunohistochemical staining on the same set of tissue samples used for KDM1A analysis, to detect ZMYM3 and HNF4A expression (Supplementary Fig. 7a). Subsequent correlation analyses unveiled a negative association between both KDM1A and ZMYM3 with HNF4A expression in HCC samples (Supplementary Fig. 7b). Additionally, we analyzed the clinical correlations of KDM1A expression level, ZMYM3 expression level, liver-TE-related gene expression profiles, and HNF4A downstream gene expression profiles in the TCGA and ICGC hepatocellular carcinoma expression profile datasets. The results showed that the expression levels of KDM1A and ZMYM3 were significantly negatively correlated with the expression profiles of liver-TE-related genes and HNF4A downstream genes (Fig. 7d), suggesting clinical relevance to our molecular-level findings.

### Discussion

In our study, we revealed the functional role and importance of liver-TEs in liver cancer. We identified liver-TEs that were enriched in histone marks, epigenetic regulators, and adjacent to a set of liver tissue-specific genes that negatively associate with poor prognosis. Liver-TE-associated genes are highly expressed in both liver tissue and liver cancer. The liver-TE-associated genes include known liver cancer suppressor genes, such as *HNF4A*[34]. Specifically, we focused on the regulatory role of liver-TEs on the expression of the *HNF4A* gene. Through the use of CRISPR/Cas9 technology, we selectively removed liver-TEs located within the transcriptional regulatory region of the *HNF4A* gene (HNF4A-liver-TEs) in liver cancer cells. Remarkably, the deletion of this

liver-TE led to a notable upregulation of HNF4A expression and tumor cell growth inhibition, providing compelling evidence for the inhibitory function of this type of DNA elements on HNF4A.

Our findings indicate the enrichment of both the histone marker H3K4me1 and its demethylase KDM1A at the liver-TEs, suggesting a complex regulation of chromatin states in this region. H3K4me1 is typically associated with enhancer regions, and its presence is a hallmark of active enhancers[38]. However, KDM1A is a demethylase that removes H3K4me1 marks from histones, resulting in gene repression[39,40]. The co-enrichment of both H3K4me1 and KDM1A at the liver-TEs may seem contradictory, but we propose several possible explanations for this observation.

One possible explanation is that KDM1A modulates H3K4me1 levels at the liver-TE sites to fine-tune their transcriptional regulatory activity. KDM1A has been shown to function as a transcriptional co-repressor and may act in concert with other chromatin regulators to silence gene expression. In this context, the presence of KDM1A at liver-TE could help prevent aberrant activation of nearby genes by modulating the levels of H3K4me1. Another possibility is that liver-TE represents a transitional state between active and inactive enhancers, where H3K4me1 and KDM1A coexist. In this scenario, the balance between H3K4me1 and KDM1A levels may determine whether the TE-derived CRE is active or repressed. Low levels of KDM1A may allow for H3K4me1 to persist and maintain an active state around TE-derived CRE, whereas high levels of KDM1A may result in H3K4me1 removal and subsequent silencing. Overall, the co-enrichment of H3K4me1 and KDM1A at the liver-TEs suggests a complex regulation of transactivation activity in this region, which is consistent with the high expression of KDM1A in liver cancer and the low expression of liver-TE-related genes in liver cancer compared to normal liver tissue. This provides mechanistic insight into the inherent connection between KDM1A and liver-TE-associated gene expression in liver cancer.

The application of CRISPRa within the context of HNF4A-liver-TE also reveals the significant regulatory influence of KDM1A. While CRISPRa technology is commonly employed to induce targeted gene expression, typically leading to transcriptional activation, our findings present a nuanced scenario. We observe that CRISPRa targeting of HNF4A-liver-TE results in a suppressive chromatin state. One plausible explanation for this observation is the transient exposure of the TE induced by CRIPSRa, which subsequently recruits KDM1A. Consequently, despite the intended activation by CRISPRa, the recruitment of KDM1A leads to the suppression of HNF4A expression. These results highlight the intricate interplay between CRISPRa-based manipulation and the context of target regions. Further investigations into the precise mechanisms underlying these atypical effects of CRISPRa are warranted, as they may offer valuable insights into the development of more refined gene manipulation strategies.

Our study emphasizes the critical role of KDM1A in regulating liver cancer cell growth through two mechanisms involving HNF4A. Firstly,

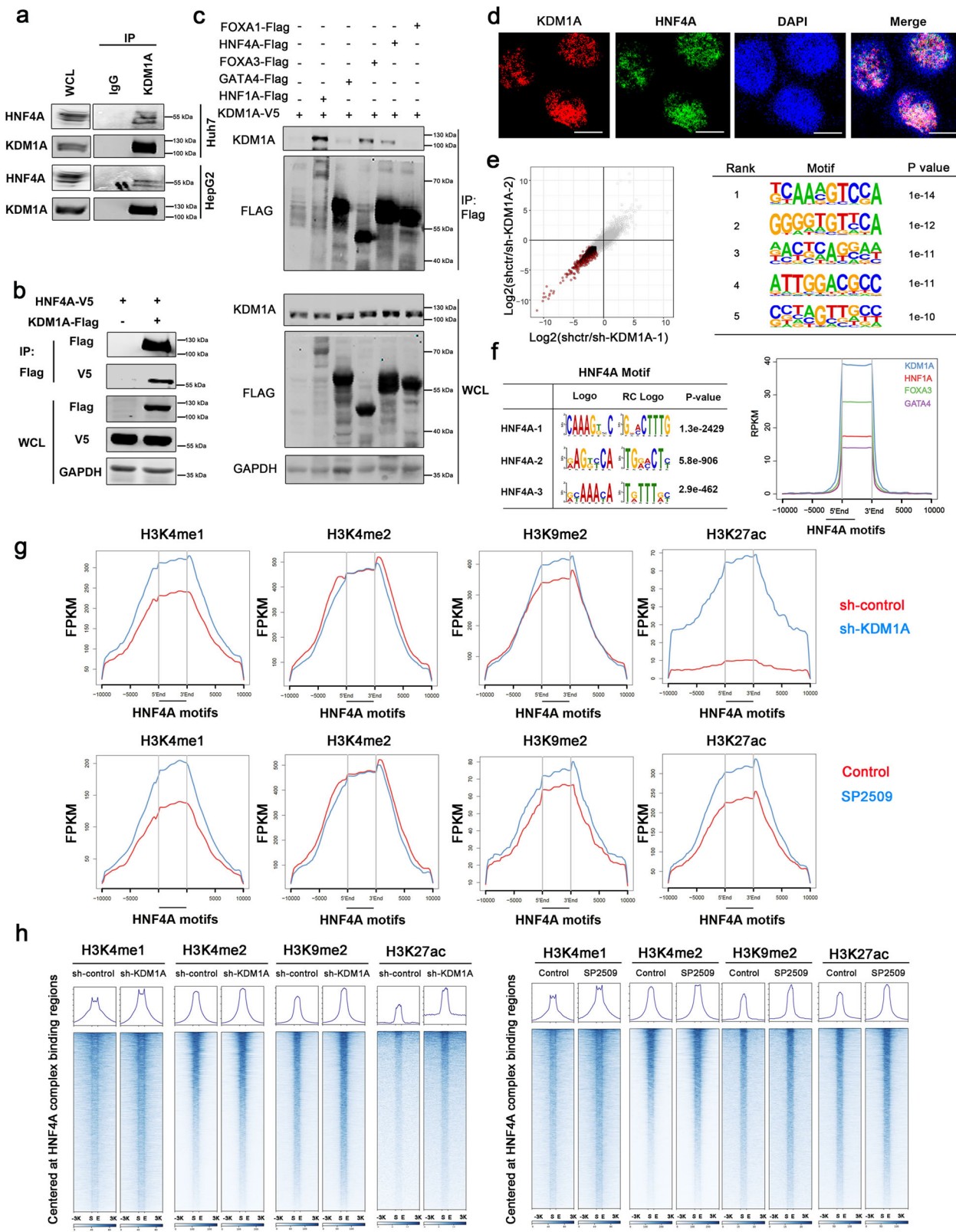

KDM1A promotes demethylation and restricts the accessibility of liver-TE, leading to the suppression of associated genes. Our study demonstrates that KDM1A directly regulates HNF4A by binding to liver-TEs within the HNF4A TRR and suppressing its expression, suggesting the involvement of KDM1A in inhibiting HNF4A expression. Secondly, KDM1A interacts with the HNF4A complex to epigenetically silence downstream genes, impacting both expression levels and

transcriptional activity of HNF4A. HNF4A emerges as a significant factor among the transcriptional regulators enriched in liver-TEs, with its target genes highly enriched in liver-TE-associated genes. This interaction and epigenetic regulation contribute to the suppression of liver-TE-associated genes by KDM1A. Our findings shed light on the regulatory mechanisms of liver cancer growth by KDM1A and HNF4A, highlighting their essential roles in liver-TE accessibility and

**Fig. 4 | Interaction between KDM1A and HNF4A regulates histone methylation.**
**a, b** Co-IP assay was performed to test the interaction between endogenous (**a**) or exogenous (**b**) KDM1A and HNF4A. Each experiment was repeated twice with similar results. **c** Co-IP assay was performed in 293T cells overexpressing V5-tagged KDM1A and one of FOXA1-Flag, HNF4A-Flag, FOXA3-Flag, GATA4-Flag, or HNF1A-Flag. Interaction of the bait protein with V5-tagged KDM1A was detected through Western blot using Flag antibody. The experiment was repeated twice with consistent results. **d** Immunofluorescence staining with KDM1A and HNF4A antibodies was performed in HepG2 cells, and their colocalization was visualized by laser confocal imaging. Bar = 25 μm. The experiment was repeated three times with similar results. **e** RNA-seq experiment was performed in stable KDM1A knockdown HepG2 cells, and genes significantly upregulated in both sh-KDM1A-1 and sh-KDM1A-2 samples (Foldchange >2, $P < 0.05$) were defined as KDM1A-repressed genes. HOMER software was used to analyze the transcription factor motifs enriched in the promoter of KDM1A-repressed genes, which utilizes the one-sided

hypergeometric test to calculate $p$-values, Benjamini–Hochberg (BH) method for $p$-value adjustment. The top 5 significant motifs were displayed. **f** Conserved recognition motifs of HNF4A were identified using MEME-ChIP software from ChIP-seq data of HepG2 cells. The significance was estimated by a one-sided Fisher's exact test, with the $p$-values multiplied by the number of discovered motifs for adjustment. The top three significantly enriched motifs are shown. ChIP-seq intensities of KDM1A, HNF1A, FOXA3, and GATA4 in the ±10 kb region of the HNF4A motifs were shown by ngsplot. **g** CUT&Tag-seq experiments of H3K4me1, H3K4me2, H3K9me2, and H3K27ac were performed in HepG2 cells with stable KDM1A knockdown or treated with SP2509. Ngsplot displays the sequencing signal intensity distribution in the ±10 kb region of the HNF4A motif for the above experiments. **h** Heatmap shows the sequencing signal intensity in the HNF4A protein complex common binding region and ±10 kb region for the above sequencing experiments. Source data are provided as a Source Data file.

transcriptional activity. The liver-TE/KDM1A/HNF4A regulatory loop exemplifies the intricate interplay between these factors in liver cancer development and progression.

We hypothesized that KDM1A requires an adapter protein with DNA-binding activity to recognize liver-TEs, as KDM1A lacks a DNA-binding domain. To test this hypothesis, we identified that ZMYM3, a zinc-finger protein with a DNA-binding domain, interacts with KDM1A. Subsequently, we demonstrated that ZMYM3 is required for the DNA-binding ability and targeted demethylation activity of KDM1A. Functionally, we found that knocking down ZMYM3 inhibits KDM1A-mediated growth of liver cancer cells. Our findings strongly suggest that ZMYM3 is necessary for the epigenetic regulatory role of KDM1A and is vital for KDM1A to exert its pro-growth effect in liver cancer cells. Previous studies have identified ZMYM3 as a subunit of the CoREST complex, but its precise role in epigenetic regulation has remained unclear[41,42]. Our results indicated the functional significance of ZMYM3, highlighting its ability to enhance the DNA-binding properties of KDM1A and direct its regulatory function to specific target sites. This mechanism of action is consistent with the broader role of protein complexes in regulating gene expression, where the specific composition of a complex determines its ability to bind to specific DNA sequences and modulate the activity of associated enzymes.

The research showed that high expression of KDM1A is associated with a poor prognosis in hepatocellular carcinoma patients, indicating its potential as a prognostic biomarker. The expression levels of KDM1A and ZMYM3 were negatively correlated with the expression profiles of liver-TE-related genes and HNF4A downstream genes, underscoring the clinical relevance of the molecular-level findings. Knockdown of KDM1A significantly inhibited liver cancer cell growth both in vitro and in vivo, highlighting the essential role of KDM1A in HCC cell growth. Additionally, the use of a small-molecule inhibitor targeting KDM1A substantially suppressed the growth of liver cancer cells, indicating KDM1A as a promising target for HCC treatment. The use of KDM1A small-molecule inhibitors could be a viable strategy for HCC treatment, and further studies are necessary to explore the clinical application of KDM1A as a prognostic biomarker and therapeutic target in HCC.

In summary, our study provides insights into the potential role of liver-TEs in modulating gene expression in liver cancer. Furthermore, our findings highlight the critical function of KDM1A in the growth of liver cancer cells through liver-TE-mediated mechanisms, specifically by repressing the transcriptional activity of HNF4A (Fig. 8). Overall, our research enhances the understanding of the precise regulation of transactivation activity at the liver-TEs, thereby opening avenues for the identification of novel therapeutic targets for liver cancer.

## Methods
### Ethical statement
Our research complies with all relevant ethical regulations. This study was approved by the Ethics Committee of Renji Hospital, Shanghai Jiao

Tong University School of Medicine, and Institutional Animal Care and Shanghai Jiao Tong University Animal Care Commission.

### Cell lines and tissue specimens
PLC/PRF/5 (CRL-8024), HepG2 (HB-8065), THLE2 (CRL-2706), THLE3 (CRL-3583), and HEK293T (CRL-3216) cells were acquired from the American Type Culture Collection (ATCC, Manassas, VA, USA). Huh7 (SCSP-526) cells were obtained from the Cell Bank of the Chinese Academy of Sciences (Shanghai, China). MHCC97H (97H) cells were provided by the Liver Cancer Institute of Zhongshan Hospital, Fudan University (Shanghai, China). Cell lines were tested by short tandem repeat (STR) profiling to verify authentication. All cells were cultured at 37 °C in 5% $CO_2$ in Dulbecco's modified Eagle medium supplemented with 10% fetal bovine serum. All cells were routinely tested using a mycoplasma detection kit (C0301S, Beytime) to ensure no mycoplasma contamination. A set of commercial tissue microarrays (TMAs) containing 90 HCC tissues and non-tumoral adjacent liver tissues were used for IHC staining. The tissues were sourced from 90 HCC patients, comprising 80 males and 10 females, with ages ranging from 16 to 73 years (median age = 48.5 years). This study was approved by the Ethics Committee of Renji Hospital, Shanghai Jiao Tong University School of Medicine.

### Animal studies
BALB/c nude mice (6 weeks old) were obtained from SLAC (Shanghai, China) and cultured under pathogen-free conditions in Shanghai Jiao Tong University Laboratory Animal Center. The mice were housed in a controlled environment with a 12-h light/dark cycle. The ambient temperature was maintained within the range of 20–26 °C, while humidity levels were kept between 40% and 70%. Daily welfare monitoring was conducted to assess their health, behavior, and any signs of distress, ensuring their well-being throughout the study period. All animal experiments were conducted in accordance with the guidelines approved by Institutional Animal Care and Shanghai Jiao Tong University Animal Care Commission. The maximal tumor size/burden permitted is 2000 mm³. To ensure humane endpoints, mice were euthanized using carbon dioxide ($CO_2$) asphyxiation followed by cervical dislocation to ensure death.

### In vivo tumorigenesis assay
For in vivo tumorigenesis assays, $2 × 10^6$ Huh7 cells expressing either vector control or specified shRNA was subcutaneously injected into male nude mice. At the end of the study, tumors were excised, weighed, and photographed.

### CRISPR/Cas9 and CRISPRa assays
For CRISPR/Cas9-mediated DNA element deletion, we used the CHOPCHOP software (http://chopchop.cbu.uib.no/) to design a pair of sgRNA flanking target elements. Then, the pair of sgRNA was

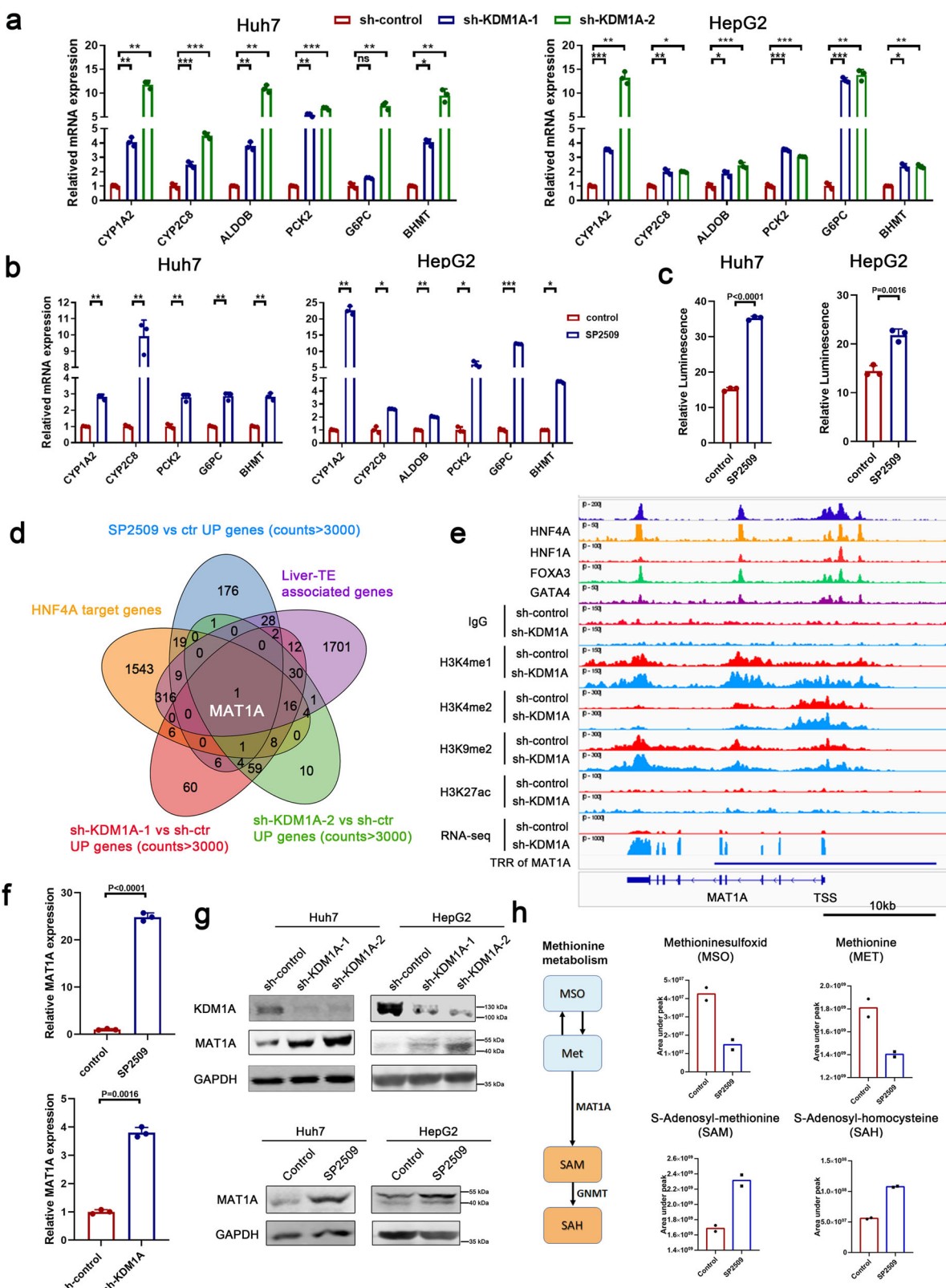

constructed into a U6-promoter-sgRNA-U6-promoter-sgRNA cassette and cloned into lentiCRISPR_v2 (Addgene #60954) for a single vector expressing two sgRNAs. The sgRNA target sequences used for DNA editing include: HNF4A-liver-TE-UP-1-sgRNA: GTTAGACAAAACTTCTC CAA(TGG); HNF4A-liver-TE-UP-2-sgRNA: GGATGCATACCCTTGGCTCC (TGG); HNF4A-liver-TE-UP-3-sgRNA: CCATCTCTCACTGGCATCCC (TGG); HNF4A-liver-TE-UP-4-sgRNA: TAAAACAGCTGCATATCCAG

(TGG); HNF4A-liver-TE-DOWN-1-sgRNA: GAGTGTTGTGTGGCCCCAC-G(AGG); HNF4A-liver-TE-DOWN-2-sgRNA: AGTGGAGGGGGCTGCACT CC(TGG); HNF4A-liver-TE-DOWN-3-sgRNA: GAAAGATCTGGGCTCAAA TC(CGG); and HNF4A-liver-TE-DOWN-4-sgRNA: GGGTGCTTAGCCC TGGTAAA(GGG).

For the CRISPRa assays, we employed the CRISPR-based Synergistic Activation Mediator (SAM) system[43], which involves two

**Fig. 5 | KDM1A negatively regulates HNF4A target genes to attenuate HNF4A function. a, b** Real-time PCR experiments were performed to analyze the expression of classical HNF4A downstream genes in KDM1A knockdown (**a**) or SP2509-treated (**b**) Huh7 and HepG2 cells, n = 3 biological replicates. Significance was examined by a two-sided t-test, and mean ± SD was shown, *$P < 0.05$, **$P < 0.01$, ***$P < 0.001$, ***$P < 0.001$. **c** HNF4A-luciferase plasmid and Renilla plasmid (control) were transfected into Huh7 and HepG2 cells treated with SP2509. Firefly and Renilla luciferase activities were measured and the ratio was used to reflect the transcriptional activation activity of HNF4A, n = 3 biological replicates. Significance was examined by a two-sided t-test, and mean ± SD was shown. **d** Venn diagram showing liver-TE-related HNF4A target genes that were significantly upregulated in KDM1A knockdown and SP2509-treated cells (sh-KDM1A-1 vs sh-control, sh-KDM1A-2 vs sh-control and SP2509 vs vehicle foldchange >2, counts >3000). **e** IGV browser view showing the binding of KDM1A, HNF4A, HNF1A, FOXA3, and GATA4 at the

transcriptional regulatory region of the *MAT1A* gene, and the effect of KDM1A knockdown on the histone modifications in the transcriptional regulatory region of the *MAT1A* gene. **f** Real-time PCR experiments were performed to detect the expression of MAT1A in KDM1A knockdown or SP2509-treated HepG2 cells, n = 3 biological replicates. Significance was examined by a two-sided t-test, and mean ± SD was shown. **g** Western blotting was performed to detect the upregulation of MAT1A in KDM1A knockdown or SP2509-treated liver cancer cells. Each experiment was repeated three times with similar results. **h** Left: Cartoon plot shows the upstream and downstream metabolites of the metabolic process catalyzed by MAT1A. Right: Metabolomics analysis was performed by mass spectrometry on control and SP2509-treated HepG2 cells. The Area under the peak value of metabolites related to methionine metabolism was determined using MS-DIAL software. n = 2 biological replicates, the bar plot shows the mean value. Source data are provided as a Source Data file.

lentiviral plasmids, one carrying dCas9-VP64 (lenti dCas9-VP64_Blast, addgene: #61425), and the other carrying the MS2-P65-HSF1 and gRNAs expressing cascade (pXPR_502, addgene: #96923). Lentivirus was generated and transduced into target cells, which were then selected with hygromycin, blasticidin, or puromycin. We targeted HNF4A-associated liver-TEs by CRISPRa system, using the following gRNA sequences: AAGGGTAGCCCTGGAGTTAG(AGG).

## Identification of liver-TEs

To identify liver-TEs, we obtained RepeatMasker-annotated transposable elements from the UCSC database. We also obtained gene annotation files (gencode.v41) for the hg38 genome from the UCSC database to extract transcription start sites (TSS). We defined transcriptional regulatory regions (TRR) as genomic regions 10 kb upstream and downstream of TSS with a Smith-Waterman score >100. We then restricted TEs for analysis to major TE class/families, including SINE (Alu, MIR), LINE (L1, L2), LTR (ERV1, ERVL, ERVL-MaLR), and DNA (hAT-Charlie, TcMar-Mariner) that overlapped with TRR (TRR-TEs).

To assess chromatin accessibility in TRR-TEs across different tissue types, we downloaded ATAC-seq data (bigWig format) from the NIH GDC data portal (https://gdc.cancer.gov/about-data/publications/ATACseq-AWG)[44]. We used the multiBigwigSummary program (parameters: BED file, −binSize 500) in the deeptools package (3.5.0) to generate the chromatin accessibility landscape, and analyzed RawCounts files using the Rtsne (0.15) to generate a tSNE map. We identified tumor type-specific TE markers using the FindConservedMarkers function, a program embedded in the Seurat package (v4.0.2) and limma (3.44.3) package. We used adj.$p$ value < 0.0001 & log2FC > 10 as the cut-off values for the FindConservedMarkers algorithm and adj.$p$ value < 0.0001 & logFC >2 and B value > 10 for the limma algorithm. We considered TEs filtered by both algorithms as liver-specific accessible TEs (liver-TEs). We defined the TRR regions containing liver-TEs as liver-TE-TRRs. To identify these regions, we utilized the "bedtools intersect" function within the bedtools software (v2.31.1), analyzing bed files containing liver-TEs and TRR regions to get TRRs harboring liver-TEs. Subsequently, the resultant bed files containing liver-TE-TRR regions were processed using the "bedtools merge" function to create a list of non-overlapping liver-TE-TRRs. In this study, we defined liver-TE-associated genes as genes that have at least one liver-TE located within 10 kb of their transcription start site (TSS). We created a gene set comprising all liver-TE-associated genes identified in our analysis. This gene set was subjected to gene set variation analysis (GSVA) and gene set enrichment analysis (GSEA) based on publicly available HCC transcriptome datasets or RNA-seq results generated from our own experiments.

The ATAC-seq data of non-tumoral tissues were obtained from BioProject PRJNA63443[45] (https://www.ncbi.nlm.nih.gov/bioproject/PRJNA63443). Samples used in this study are listed in Supplementary Data 7. The uniquely mapped reads were isolated by using samtools (v1.6) under parameters: samtools view -f 2 -q 10 -b. Then, the Bam files

including uniquely aligned reads were transformed to bigwig formatted files by the bamCoverage function in Deeptools (v3.5.1) with default parameters. Then, we used the multiBigwigSummary program (parameters: BED file, −binSize 500) in the deeptools package (3.5.0) to analyze the chromatin accessibility landscape.

## Reagents and antibodies

SP2509 (S7680), Seclidemstat (S6722), GSK-LSD1 (S7574), ORY-1001 (S7795), and GSK2879552 (S7796) were obtained from Selleck. Puromycin and penicillin/streptomycin were purchased from yeason, China. The primary antibody were as followed: anti-KDM1A (abcam, ab129195[EPR6825], lot1018822-16, 1:100 for CUT&Tag, IF and IHC, 1:1000 for Western blots; ABclonal, A21801[O60341], lot3560844003, 1:200 for IP and 1:1000 for Western blots); anti-ZMYM3 (Proteintech, 25742-1-AP[AB_2880221], lotC10-027S, 1:100 for IHC, 1:200 for IP and 1:1000 for Western blots); anti-HNF4A (R&D, PP-H1415-00[Cl H1415], lotA-2, 1:200 for IF, 1:100 for IHC and 1:1000 for Western blots; ABclonal, A20865[ARC2794], lot3560844003, 1:1000); anti-H3K4me1 (abcam, ab176877[ERP16597], lotGR3208750-3, 1:100 for CUT&Tag and 1:5000 for Western blots); anti-H3K4me2 (abcam, ab32356[Y47], lotGR253788-33, 1:100 for CUT&Tag and 1:5000 for Western blots); anti-H3K27ac (abcam, ab4729, lot1059037-1, 1:100 for CUT&Tag); anti-H3K9me2 (abcam, ab1220[mAbcam 1220], 1:100 for CUT&Tag and 1:5000 for Western blots), anti-FLAG (ABclonal, AE063[ARC5111-02], lot9100026002, 1:200 for IP and 1:5000 for Western blots); anti-MAT1A (ABclonal, A2630[AB_2764502], lot002800101, 1:1000); anti-V5 (ABclonal, AE017[AMC0506], lot9200017002, 1:200 for IP and 1:5000 for Western blots); anti-GAPDH (Santa Cruz biotechnology, sc-32233[6C5], lotG3020, 1:5000); IP lysis buffer (Beyotime Institute of Biotechnology, P0013); RIPA (Beyotime Biotechnology, P0013B) ; Flag M2 affinity gel (Bimake, B26101), Protein G Dynabeads (Thermo Fisher, 10017D); Poly Flag peptides (Bimake, Poly FLAG F4799). Second antibody for Western blot: Goat anti-rabbit (LI-COR, 926–3221, lotD31205-5, 1:10,000); Goat anti-mouse (LI-COR, 926–68020, lotD00310-25,1:10000). Second antibody for Immunofluorescence analysis: Goat anti-rabbit (Thermo Fisher, A11008, lot2420731, 1:1000); Goat anti-mouse (Thermo Fisher, A11020, lot2306811, 1:1000); DAPI (Thermo Fisher, D1306, 1:1000).

## Plasmids and stable cell line construction

The ORF of KDM1A (shRNA-resistant, wild-type, or K661A mutant) were cloned into pLVX-IRES-Puro or pLVX-IRES-ZsCreen plasmid. The specific shRNA sequence was designed and inserted into pLKO.1: shKDM1A(sh1: CCGGGCTACATCTTACCTTAGTCATCTCGAGATGACTA AGGTAAGATGTAGCTTTTTG;   sh2:CCGGGCCTAGACATTAAACTGAA-TACTCGAGTATTCAGTTTAATG   TCTAGGCTTTTTG);shHNF4A(CCGG TCAGGGTCTGAGCCCTATAAGCTCGAGCTTATAGGGCTCAGACCCTG ATTTTTG);shZMYM3(CCGGGTTGTACCGGGCTCAACTATTCTCGA-GAATAGTTGAGCCCGGTACAACTTTTTG). HEK293T cells and packaging plasmids (psPAX2, pMD2.G) were used to produce lentivirus. HCC

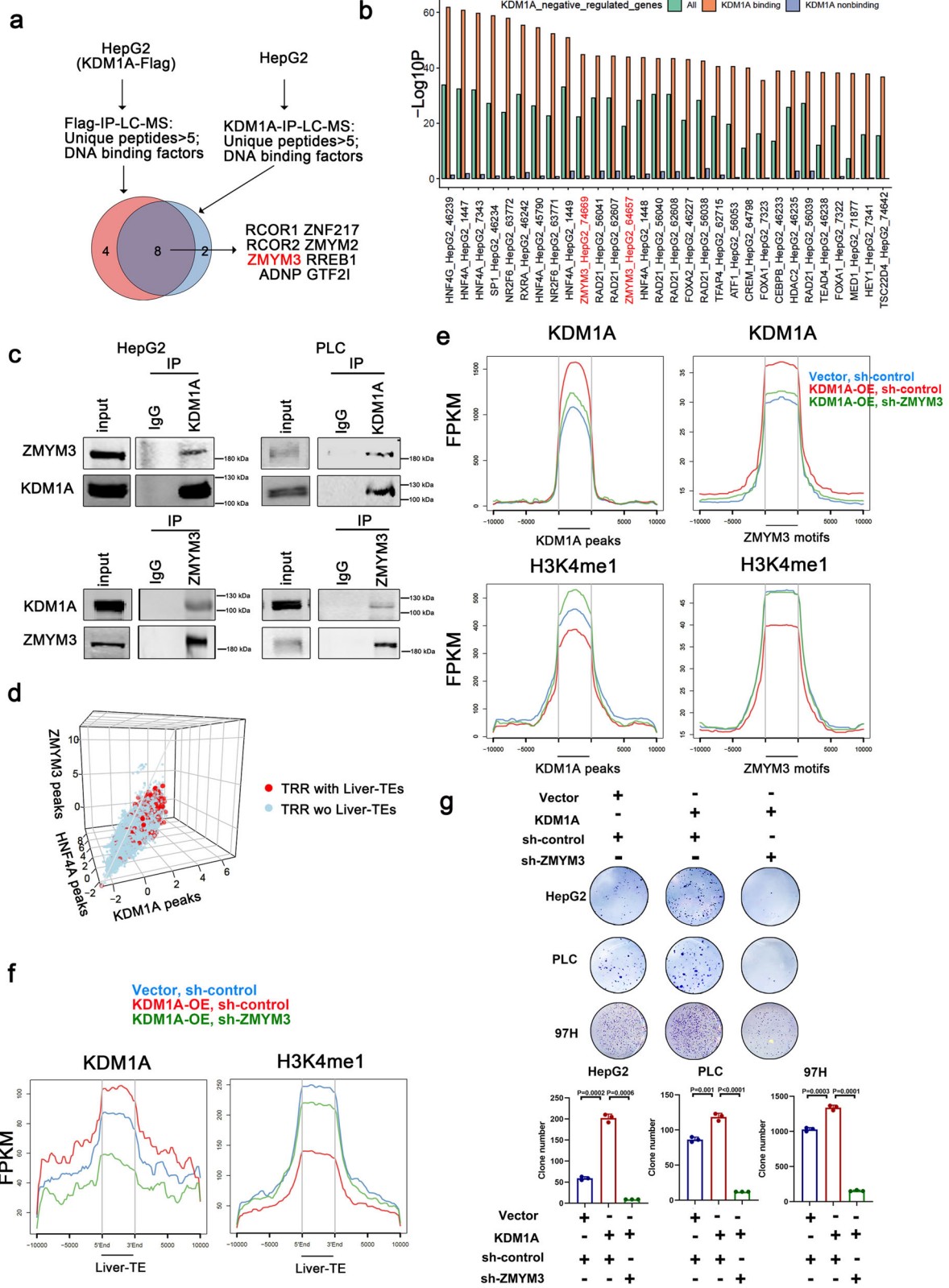

cells were infected with lentivirus in the presence of polybrene. Cells were collected for gene expression assay.

### Identification of transcriptional regulators and histone markers enrichment in liver-TEs

To determine the enrichment of transcription regulatory (TR) proteins or histone markers (HM) in liver-TEs, we utilized the Remapenrich

package (0.99.0) (https://github.com/remap-cisreg/ReMapEnrich). First, we obtained a bed-formatted file containing the positions of liver-TEs as the query set. For TR enrichment analysis, we downloaded the bed-formatted Remap catalog (2020 version) from the Remap database (https://remap.univ-amu.fr/storage/remap2020/hg38/MACS2/remap2020_all_macs2_hg38_v1_0.bed.gz)[46]. We then extracted ChIP-peaks from liver cancer HepG2 cells to assemble a sub-catalog for

**Fig. 6 | ZMYM3 is required for the epigenetic regulatory role of KDM1A in liver cancer cells.** (**a**) Screening KDM1A interacting proteins using LC-MS. **b** KDM1A negatively regulated genes were screened from RNA-seq data of HepG2 cells with KDM1A knockdown (sh-KDM1A-1 vs. sh-control and sh-KDM1A-2 vs. sh-control, Foldchange >2 and $P < 0.05$), and then genes bound by KDM1A were selected based on HepG2 cell KDM1A-ChIP-seq data. Finally, the transcriptional regulatory regions of genes directly negatively regulated by KDM1A were analyzed for transcriptional regulator (TR) enrichment using LISA software. The TRs that significantly enriched for KDM1A bound and negatively regulated genes were ranked by −Log$P$ values and shown. LISA utilizes a one-sided Wilcoxon rank sum test to calculate $p$-values without adjustment. **c** Co-IP assay was performed in HepG2 and PLC cells using endogenous KDM1A or ZMYM3 as bait protein to verify the interaction between ZMYM3 and HNF4A. Each experiment was repeated three times with similar results. **d** The 3D plot shows the correlation among the ChIP-seq signal intensities of KDM1A, ZMYM3, and HNF4A at the gene transcription regulatory regions (TSS ± 10 kb). The Spearman correlation coefficients between each pair of these proteins

were shown in Supplementary Fig. 6b. Red and blue points represent the TRR region with and without liver-TE, and the comparisons between peak scores in these two types of TRR were shown in Supplementary Fig. 6c. **e, f** CUT&Tag-seq analysis reveals the effects of ZMYM3 knockdown on KDM1A binding and H3K4me1 modification. CUT&Tag-seq experiments were performed using KDM1A and H3K4me1 antibodies in three groups of 97H cells: Vector control + sh-control, KDM1A + sh-control, and KDM1A + sh-ZMYM3. Ngsplot was used to display the changes in KDM1A binding strength and H3K4me1 modification within ±10 kb regions surrounding KDM1A binding sites, ZMYM3 motifs (**e**), and liver-TEs (**f**). The results showed that the knockdown of ZMYM3 decreased the binding of KDM1A to its target regions and increased the corresponding H3K4me1 modifications. **g** Colony formation ability was assessed in HepG2, PLC, and 97H cells across three groups: Vector control + sh-control, KDM1A + sh-control, and KDM1A + sh-ZMYM3, n = 3 biological replicates. Significance was examined by a two-sided t-test, and mean ± SD was shown. Source data are provided as a Source Data file.

this analysis. For HM enrichment analysis, we downloaded the HM ChIP-seq peaks of HepG2 cells from the GTRD database (http://gtrd.biouml.org:8888/downloads/20.06/bigBeds/hg38/ChIP-seq_HM/Peaks/) and assembled a custom Remap-catalog formatted file. For both TR and HM enrichment analyses, the software generated background regions by randomly shuffling the query regions within TRR for 500 iterations (shuffles=500). Other parameters in the program were set as follows: byChrom = F, fractionQuery = 0.01, fractionCatalog = 0.01, included = 0.9, tail = "both". The enrichment of TR or HM in liver-TEs was then determined using the Remapenrich package.

To analyze the TRs that enriched the promoters of a selected set of genes, Lisa software (https://github.com/liulab-dfci/lisa) was also used. It uses a gene-centric approach to identify transcription factors and other regulatory proteins by integrating online ChIP-seq data.

## Western blot and co-immunoprecipitation (Co-IP)

To perform Western Blot, the cells were lysed with RIPA buffer (Beyotime Biotechnology, P0013B) containing a protease inhibitor cocktail (bimake, B14001) for 1 h on ice. Next, the cell lysates were centrifuged at $13,000 \times g$ for 20 min at 4 °C. The protein concentrations of the supernatants were determined using a BCA protein assay kit (Beyotime Biotechnology, P0011). To separate the protein samples, 100 μg total proteins were loaded onto SDS-PAGE and transferred to nitrocellulose (NC) membranes (Pall Corporation). The NC membranes were then blocked with 5% BSA and incubated with primary antibodies overnight at 4 °C. After washing the membranes three times with TBST, they were incubated with goat anti-rabbit and goat anti-mouse antibodies and visualized using the Odyssey imaging system.

For Co-immunoprecipitation (Co-IP), the cells were lysed with IP lysis buffer containing protease inhibitor cocktail for 1 h on ice. The supernatant proteins were then incubated with 50 μl of Protein G Dynabeads (Thermo Fisher, 10017D) on a rocking platform for 3 h at 4 °C to reduce non-specific binding. After removing the Dynabeads, the supernatants were incubated with 50 μl of Protein G Dynabeads and primary antibody on a rocking platform overnight at 4 °C. The immunoprecipitates were collected using a magnet and washed with TBS three times. The Dynabeads were denatured with a loading buffer and subjected to Western Blot analysis. For Co-IP conducted with Flag M2 affinity gel (Bimake, B26101), the supernatant proteins were incubated with 50 μl of Flag M2 affinity gel on a rocking platform overnight at 4 °C. The subsequent steps were similar to those of Co-IP conducted with Protein G Dynabeads.

## LC-MS protein identification

To identify KDM1A interacting proteins, protein complexes were immunoprecipitated from HepG2 cells expressing KDM1A-Flag using an anti-Flag antibody (n = 1). Simultaneously, protein complexes were also immunoprecipitated from HepG2 cells using an anti-KDM1A

antibody (n = 1). In both cases, an anti-IgG antibody was used as a control. Subsequently, the immunoprecipitated KDM1A complexes were subjected to LC-MS identification. The purified protein complex was resolved by SDS-PAGE followed by excised, digested, and then subjected to LC-MS (Liquid chromatography-tandem mass spectrometry) analysis. This approach was accomplished by PANOMIX Bio-Medical Tech. Co. Suzhou, China.

Simple preparation: The gel is cut into small pieces and treated with a destaining solution containing 40% methanol and 50 mM $NH_4HCO_3$. The gel pieces are heated and shaken for 15 min, with the destaining solution replaced at least once. After destaining, the gel pieces are washed twice with water and dehydrated with 75% ACN. The gel pieces are then washed with $NH_4HCO_3$ and subjected to protein digestion. To digest the proteins, $NH_4HCO_3$ is removed, and trypsin is added to the gel pieces. The gel pieces are ground and centrifuged, and the mixture is incubated overnight at 37 °C. The resulting peptides are extracted with ACN, centrifuged, and the supernatant is collected and dried.

Liquid chromatography (LC) was performed using an Easy-nLC 1200 system (Thermo Fisher Scientific, USA). Samples were dissolved in 80 μL of mobile phase A (0.1% formic acid aqueous solution), vigorously mixed, and then 5 μL of the sample was loaded into a 20 μL quantitative loop using an autosampler. The sample in the quantitative loop was loaded onto a C18 reverse-phase pre-column by pump A at a maximum pressure of 280 bar. Subsequently, the sample was eluted onto a C18 analytical column using a gradient of 11%−37% mobile phase B (0.1% formic acid in 80% acetonitrile) at a flow rate of 600 nl/min for separation.

Mass spectrometry data were acquired using an Orbitrap Fusion Lumos mass spectrometer (Thermo Fisher Scientific, USA). The ion source employed was a nanospray ionization source (NSI) with a spray voltage of 2200 V and an ion transfer capillary temperature of 320 °C. Mass spectrometric data were collected in data-dependent acquisition (DDA) mode under positive ion mode. In the first stage, full scans were performed using the Orbitrap with a scan range of m/z 350−1550, at a resolution of 120,000, with automatic gain control (AGC) set to 5e5 ions, and a maximum injection time of 50 ms. For the second stage, fragmentation of precursor ions was achieved by high-energy collision dissociation (HCD) at 32%, with fragment ions detected in the Orbitrap at a resolution of 15,000. The first mass was set to 100, with an AGC target of 5e4 ions and a maximum injection time of 22 ms. A precursor ion selection window of 1.6 Th was employed, and MS/MS acquisition was performed for ions with charge states ranging from 2 to 7. Dynamic exclusion was set to exclude the same precursor ion from MS/MS acquisition for 30 s after being selected once for MS/MS acquisition.

The Sequest algorithm integrated in Proteome Discoverer software (v2.2) was used for database searching. PD software was used for qualitative analysis based on Sequest search results and the first-step spectrum screening. Protein quantification values were the sum of the

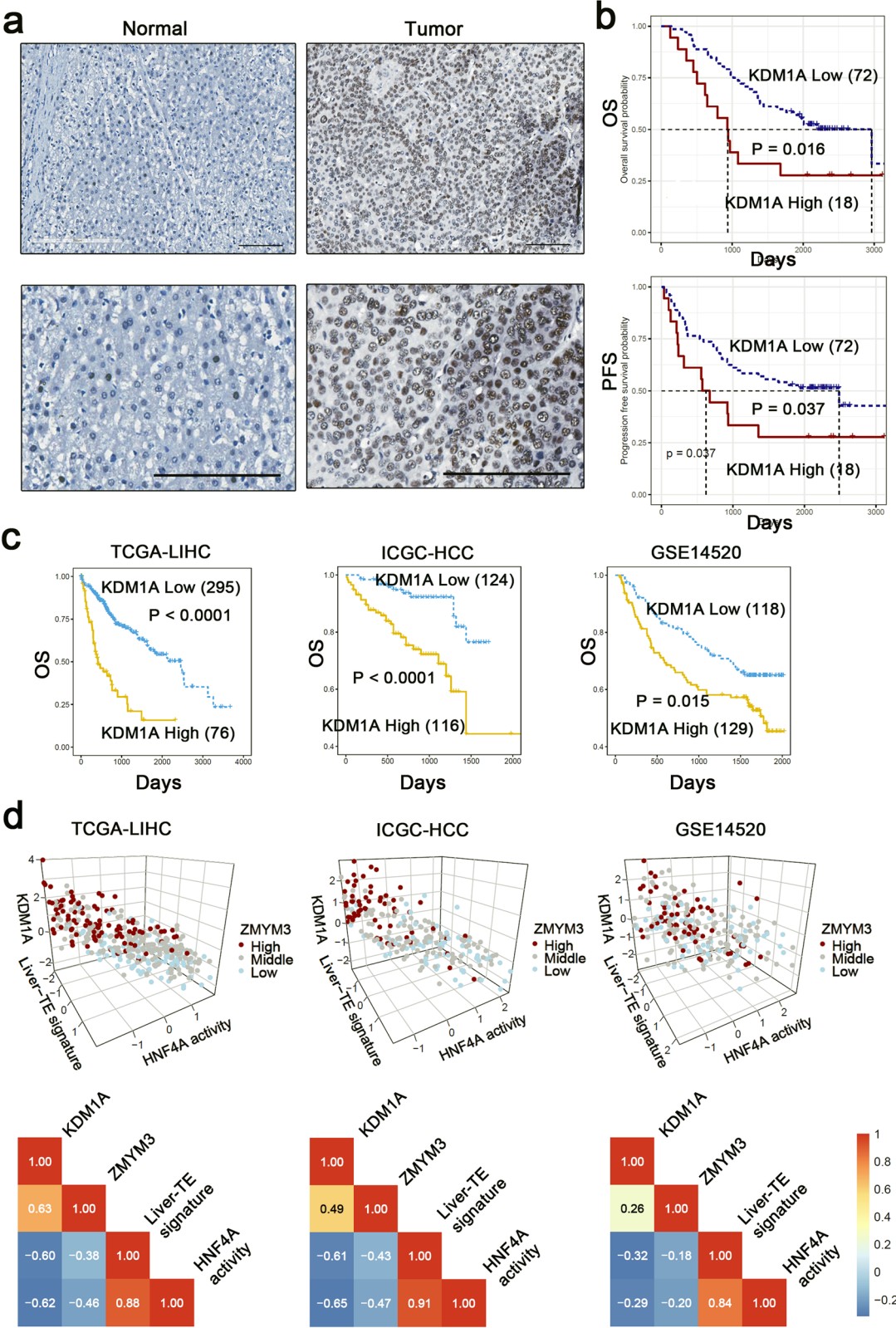

peptide quantification values. Search parameters were as follows: MS tol: 20 ppm, MS/MS tol: 0.05 Da, max missed cleavages: 2. The database used was Uniprot (Human, version 202002).

**Quantitative real-time PCR (RT-PCR)**
Total RNA was extracted with the RNAiso Plus kit (Takara Bio Inc) according to the manufacturer's instructions. RNA was reverse transcribed to cDNA with reverse transcriptase (yeason, 14605ES08). Quantitative real-time PCR (RT-PCR) was performed with qPCR SYBR Green Master Mix (yeason, 11199ES03) and analyzed by Viia7 Real-Time PCR System (AB Applied Biosystems). The sequences of primers used for RT-PCR are provided in Supplementary Data 8. Data are normalized to GAPDH expression in each sample.

**Fig. 7 | KDM1A is overexpressed in hepatocellular carcinoma and predicts poor prognosis. a** Immunohistochemical staining of KDM1A was performed in 90 pairs of hepatocellular carcinoma tissues and adjacent non-tumoral liver tissues. The representative immunohistochemical staining images of KDM1A were shown (scale bar = 100 μm). **b** Kaplan–Meier curves showing the relationship between KDM1A expression levels and overall survival (OS)/progression-free survival (PFS). Protein expression high/low groups were determined by the Maxstat R program. The difference between groups was tested by a two-sided Log-rank test. n = 90 samples, High = 18, Low = 72. **c** Kaplan–Meier curves showing the prognostic (OS) relevance

of KDM1A in TCGA (n = 371 samples, High = 76, Low = 295), ICGC (n = 240 samples, High = 116, Low = 124), and GSE14520 (n = 247 samples, High = 129, Low = 118) liver cancer expression profile datasets. **d** Pearson correlation analysis of KDM1A/ ZMYM3 expression levels, liver-TE-related gene expression profiles, and HNF4A downstream gene expression profiles in TCGA, ICGC, and GSE14520 liver cancer datasets. The 3D plot and heatmap display the Pearson correlations of the above values in liver cancer samples. All P-values are significant (P < 0.001). Source data are provided as a Source Data file.

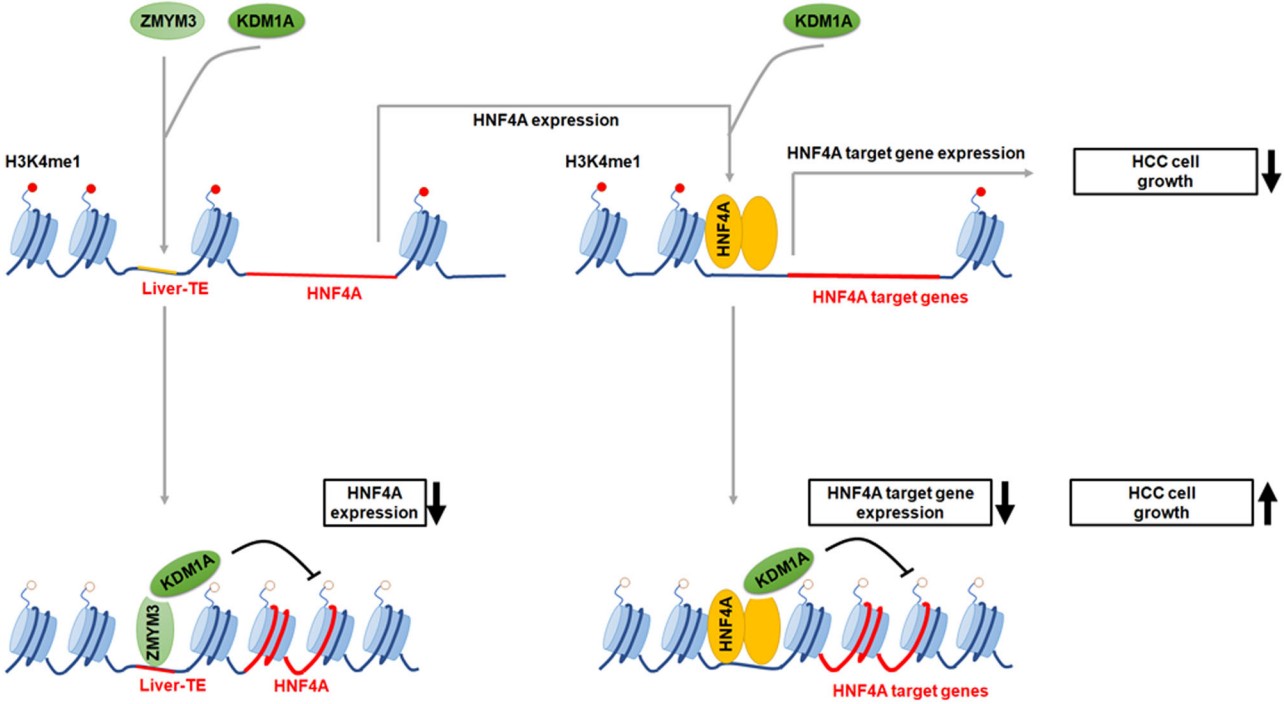

**Fig. 8 | Schematic diagram illustrating the regulatory mechanism of liver-TEs in liver cancer cell proliferation.** In HCC cells, liver-TEs exhibit high plasticity and regulatory potential, and they are located adjacent to several tumor-suppressive genes. liver-TEs are enriched with motifs recognized by ZMYM3, which recruits the histone demethylase KDM1A. This study focuses on liver-TE-associated genes and the regulatory mechanisms involved in the expression of HNF4A, an important tumor-suppressive gene in HCC. The following mechanisms were illustrated: (1) KDM1A/ZMYM3 recruitment to the transcriptional regulatory region of the *HNF4A* gene through liver-TE, leading to specific removal of the active histone mark H3K4me1 by KDM1A; (2) KDM1A interacts with HNF4A, thereby suppressing the expression of downstream genes; and (3) liver-TE/ZMYM3/KDM1A promotes HCC cell growth by inhibiting the expression and activity of HNF4A.

## Immunofluorescence analysis

Cells were seeded on coverslips and cultured for 24 h. Then cells were fixed with methanol and permeabilized with 0.1% Triton X-100 for 10 min. Cells were washed with PBS three times and a blocking buffer (5% goat serum) was applied for 30 min at room temperature. The cells were incubated with the primary antibody overnight at 4 °C. After washing three times in PBS, the cells were incubated with a second antibody for Immunofluorescence analysis for 1 h. Cell nuclei were counterstained using DAPI. The slides were observed under a Leica SP8 confocal microscope (Leica Microsystems, Wetzlar, Germany).

## Immunohistochemistry

The tissue microarrays underwent deparaffinization, rehydration, immersion in 3% hydrogen peroxide solution for 15 min, antigen retrieval in citrate buffer (pH 6.0) for 25 min at 95 °C, and cooling for at least 60 min at room temperature. They were then blocked with 10% normal goat serum for 30 min. Next, the slides were incubated with primary antibody overnight at 4 °C and visualized using the PV-9000 Polymer Detection System (GBI, USA) following the manufacturer's instructions. After washing with PBS, the slides were counterstained with hematoxylin. Protein expression levels were determined based on staining intensity and percentage of immunoreactive cells, where

staining intensity was graded as 0 (negative), 1 (weak positive), 2 (moderate positive), and 3 (strong positive), and immune reaction cell percentage were classified as 0 (0%), 0.5 (1-10%), 1 (11-20%), 2 (21-50%), 3 (51-80%), 4 (81-100%). The final score for a sample was calculated as the mean of tumor cell staining intensity scores multiplied by the percentage of positive cell scores. The prognostic value of certain proteins in patients with different subtypes was evaluated using the R software packages survival (3.2-13) and survminer (0.4.9). The best cut-off value for the IHC staining score was estimated using the R package maxstat (0.7-25).

## Cell viability assays

In the colony formation assays, cells were placed in 6-well plates and allowed to incubate for a period of 14 days. The resulting colonies were stained with crystal violet (Beyotime, C0121) for visualization. Each well was seeded with 1000 cells for gain of function assays, 2000 cells for loss of function assays, and 10,000 cells for in vitro pharmacological assays.

## Luciferase reporter assay

HNF4-Luc Reporter plasmid was purchased from Genomeditech. Renilla luciferase Reporter plasmid was acquired from Inovogen Tech. Co. $5.0 \times 10^4$ cells were seeded in a 12-well plate and co-transfected

with the above vectors when the cell confluence reached 80%. After 24 h, the cells were treated with a Luciferase Reporter Assay Substrate Kit (Beyotime Institute of Biotechnology, RG027) following the manufacturer's instructions, and the luciferase activity was measured using a SpectraMax reader (Molecular Devices, Shanghai, China).

## Publicly available transcriptome and epigenome datasets

The HCC dataset from The Cancer Genome Atlas (TCGA) database[47] (https://portal.gdc.cancer.gov/) was obtained using the TCGAbiolinks package (2.16.4). FPKM-normalized RNA-seq data were converted to TPM values using the TCGA dataset for further analysis. Additionally, the GSE14520[48] (https://www.ncbi.nlm.nih.gov/geo/query/acc.cgi?acc=GSE14520) and GSE54236[49] (https://www.ncbi.nlm.nih.gov/geo/query/acc.cgi?acc=GSE54236) HCC transcriptome datasets were obtained from the Gene Expression Omnibus (GEO) database using the GEOquery package (2.56.0). The HCC dataset from the International Cancer Genome Consortium (ICGC) database was obtained from the ICGC data portal (https://dcc.icgc.org/)[50]. From the UCSC Xena database (https://xenabrowser.net/), we downloaded a combined TCGA and Genotype-Tissue Expression (GTEx) dataset[51] (https://xenabrowser.net/datapages/?cohort=TCGA%20TARGET%20GTEx&removeHub=https%3A%2F%2Fxena.treehouse.gi.ucsc.edu%3A443) for analyzing the overall expression of liver-TE associated genes in different types of normal tissues/cancer tissues.

To investigate the prognostic relationship of liver-TE-associated genes, we used our created liver-TE gene signature to calculate the GSVA score of liver-TE using the GSVA package. Next, survival analyses were performed on transcriptome datasets containing survival information using the survival (3.2-11) and survminer (0.4.9) packages in R based on the GSVA score in each sample. Cut-off values were estimated using the maxstat package (0.7–25). Furthermore, the maxstat method was also utilized to stratify KDM1A expression for Kaplan–Meir survival analysis.

Several publicly available ChIP-seq raw data were downloaded from the ENCODE database (https://www.encodeproject.org/), including ENCFF750VZA (KDM1A-ChIP-seq in HepG2 cells), ENCFF000PJT (HNF4A-ChIP-seq in HepG2 cells), ENCSR848YWD (ZMYM3-ChIP-seq in HepG2 cells), ENCFF442RQA (HNF1A-ChIP-seq in HepG2 cells), ENCFF492CBJ (FOXA3-ChIP-seq in HepG2 cells), and ENCFF163SRP (GATA4-ChIP-seq in HepG2 cells)[45,52]. To determine protein binding regions based on these data, adapters were trimmed using Trim-galore (0.6.5-1), and reads were aligned to the human genome (UCSC hg38) using BWA (0.7.17). The aligned reads were then processed by MACS2 (v2.2.7.1) for peak calling, and bigwig files for peak visualization in Integrative Genomics Viewer (IGV) were generated using Deeptools (v3.5.1). For the KDM1A-ChIP-seq data, we categorized genes with KDM1A peaks (identified by MACS2, with a score >200) within a ±10 kb vicinity of their TSS as KDM1A target genes. To analyze the methylation status of CpG sites in liver-TEs, we downloaded a BED file (GSM1204463[53], https://www.ncbi.nlm.nih.gov/geo/query/acc.cgi?acc=GSM1204463) from the GEO database. This file contained the frequency of methylated reads in each detected CpG site in HepG2 cells, which were obtained using a Reduced Representation Bisulfite Sequencing (RRBS) assay. We shifted the reference genome from hg19 to hg38 using the liftOver software and used BEDTools (v2.31.1) to extract CpG sites within liver-TEs for comparison of methylation levels with a similarly sized sample of randomly selected TEs within TRR regions.

Capture Hi-C data were obtained from the ArrayExpress database (E-MTAB-7144)[54], originating from HepG2 cells. The raw sequencing data underwent preprocessing using the HiC-Pro software (3.1.0). This included alignment to the human hg38 genome, removal of duplicate reads, filtering for valid interactions, and generation of binned interaction matrices. cis-interactions involving liver-TEs were selected using the bedtools (v2.31.1), and visualized by the IGV genome browser.

## ChIP-PCR

ChIP assays were conducted with a minimum of $1 \times 10^7$ cells per sample. The cells were initially crosslinked with 1% formaldehyde for 10 min at room temperature, and the crosslinking reaction was stopped by 0.125M glycine. Subsequently, the cells were washed with cool PBS and collected by centrifugation at $700 \times g$ for 5 min at 4 °C. Nuclei were isolated using a nuclear extraction kit (Solarbio, EX2650). The isolated nuclei were suspended in a 20% SDS lysis buffer and incubated on ice for 10 min, followed by sonicating to fragment the DNA into sizes ranging from 200 to 1000 base pairs. The resulting nuclear lysates were then clarified by centrifugation at $9400 \times g$ for 10 min at 4 °C. 1% aliquot of the supernatant was reserved for use as the whole-cell extract (WCE) DNA control, while the remaining supernatant was diluted with a dilution buffer consisting of Triton X-100, EDTA, NaCl, Tris-HCl pH 8, and a protease inhibitor. The diluted supernatant was then pre-cleared using protein G Dynabeads (Thermo Fisher, 10017D) on a rocking platform for 3 h at 4 °C to reduce non-specific binding. After removal of the Dynabeads, the supernatants were incubated overnight at 4 °C with 50 µl of Protein G Dynabeads and the primary antibody on a rocking platform. Protein/DNA immunoprecipitates were subsequently collected using a magnet and subjected to a series of washes, including a low salt wash buffer, high salt wash buffer, LiCl wash buffer, and TE buffer. Reversal of protein/DNA cross-links in both WCE DNA and immunoprecipates was carried out by Elution buffer (0.5 M NaHCO₃, SDS%) via incubating at 65 °C for 6 h. The immunoprecipitated protein and RNA were then digested with Proteinase K and RNAse A at 55 °C for 2 h. DNA Purification was carried out according to the manufacturer's instructions for the Qiaquick PCR Purification Kit (Qiagen, 28106). The antibodies used in ChIP-PCR assays were anti-Flag antibody (ABclonal, AE063, 1:100) and anti-H3K4me1 antibody (abcam, ab176877, 1:100). The Primers used in evaluating Flag-KDM1A occupancy at the HNF4A-liver-TE region are as following: Forward: GGGCCCCAAGTCTATGGTTC and Reverse: AGG-CACCCACAAAGCTTCAA. The primers used in detecting H3K4me1 levels at the HNF4A TRR regions were primer pair 1: GTTCTCCA-CAGGGAGGTAG, GGTGAGCACCTGCTGAGCTG, and primer pair 2: GCTCGGCTGACCTCAG, ACAAGCAGACACTGCCGCA.

## RNA-seq

Total RNA was extracted using TRIzol Reagent (Life Technologies, CA, USA) according to the manufacturer's protocol, and RNA integrity was assessed using an Agilent Bioanalyzer 2100 (Agilent Technologies, CA, USA) to obtain RIN values. RNA with RIN values >7 was purified using an RNAClean XP Kit (Beckman Coulter, Inc. CA, USA) and RNase-Free DNase Set (QIAGEN, GmBH, Germany). RNA libraries were prepared for sequencing using a VAHTS Universal V6 RNA-seq Library Prep Kit for Illumina (Vazyme, Nanjing, China), and sequencing was performed on an Illumina HiSeq 2500 system.

Raw RNA-seq data were subjected to quality control using FastQC (v0.11.9) and trimmed using Trim_galore (0.6.5-1). SortMeRNA (4.2.0) was used to remove rRNA reads to generate clean data. The clean reads were aligned to the human reference genome (hg38) using the STAR (2.7.6a) aligner to generate BAM files. Duplicates were removed using Samtools (1.7), and the unique mapped counts were obtained using featureCounts (2.0.1). Differential gene expression analysis was performed using EdgeR (3.30.3), and gene set enrichment analysis (GSEA) and gene function enrichment were performed using the ClusterProfiler package (3.16.1) based on the EdgeR results. The aligned reads (BAM files) were transformed into bigwig files using Deeptools software (v3.5.1) for visualization in Integrative Genomics Viewer (IGV).

## CUT&Tag-seq

We used the Hyperactive® Universal CUT&Tag Assay Kit for Illumina (Vazyme #TD903) to prepare the DNA library. Briefly, fifty thousand cells were gently resuspended with NE buffer and incubated on ice for

10 min. The nuclei were isolated and conjugated to 10 μL pre-activated concanavalin A-coated magnetic beads. The bead-bound nuclei were then incubated with 50 μL Antibody Buffer containing anti-biotin rabbit mAb (1:100 dilution) overnight at 4 °C. Subsequently, the nuclei were resuspended with 100 μL dig-wash buffer containing antibody (1:100 dilution) to bind with the primary antibody or rabbit IgG (control). The nuclei were washed three times with 200 μL of dig-wash buffer and tagmented with pA/G-Tn5 adapter complex. The tagmented DNA was collected using DNA Extract Beads for library preparation. PCR was performed for library amplification for 15 cycles, and the library was purified using VAHTS DNA Clean Beads (Vazyme #N411). The tagmented DNA was sequenced using the Illumina novaseq 6000 platform.

The sequencing reads obtained from the Illumina platform were subjected to quality control using FastQC (v0.11.9) to assess the quality of the reads. Adapter trimming and read filtering were performed using Trim-galore (0.6.5-1) with the following parameters: –phred33 –length 35 –stringency 3. The filtered reads were then aligned to the human genome (UCSC hg38) using Bowtie2 version 2.2.6 with parameters: –local –very-sensitive –no-mixed –no-discordant –phred33. The aligned reads were processed by MACS2 (v2.2.7.1) to call peaks and transformed to bigwig formatted files by the bamCoverage function in Deeptools (v3.5.1) with default parameters. To visualize the genomic tracks and peak profiles, the processed data were visualized using the Integrative Genomics Viewer (IGV), Ngsplot (v2.63) or Deeptools plotheatmap function. The differential peaks between groups were determined by the macs2 bdgdiff function with paramter: –cutoff 2.

The primary antibody used in CUT&Tag-seq assays were as followed: anti-KDM1A (abcam, ab129195, 1:100); anti-H3K4me1 (abcam, ab176877, 1:100); anti-H3K4me2 (abcam, ab32356, 1:100); anti-H3K27ac (abcam, ab4729, 1:100) and anti-H3K9me2 (abcam, ab1220, 1:100).

## ATAC-seq

The ATAC-seq library preparation was conducted using the Hyperactive ATAC-Seq Library Prep Kit for Illumina (Vazyme, TD711) according to the manufacturer's guidelines. Each sample comprised a total of $5 \times 10^5$ cells. Initially, cells were washed with 500 μl PBS and then centrifuged at $500 \times g$ for 5 min at room temperature. Subsequently, the cell pellets were resuspended in 50 μl of cold lysis buffer and incubated on ice for 10 min to isolate the nuclei. Following centrifugation at $500 \times g$ at 4 °C for 5 min, the nuclei were subjected to a transposition reaction by incubating them with a 50 μl Tn5 transposome/Transposition reaction mix at 37 °C for 30 min. The tagmentation process was performed within the transposition reaction system. The fragmented/transposed DNA was purified using VAHTS DNA Clean Beads. The purified DNA underwent two washes with 200 μl of fresh 80% ethanol and was finally eluted in 26 μl of Nuclease-free ddH$_2$O. The library amplification protocol followed the program: 72 °C for 3 min; 95 °C for 3 min; 12–15 cycles of 98 °C for 10 s, 60 °C for 5 s; 72 °C for 1 min; and then held at 12 °C. The amplified ATAC-Seq library was further purified using VAHTS DNA Clean Beads. The purified DNA was washed twice with 200 μl of fresh 80% ethanol and eluted in 22 μl of Nuclease-free ddH$_2$O. Finally, all ATAC-seq libraries were sequenced using the Illumina NovaSeq 6000 platform.

The sequencing data obtained from the Illumina platform underwent quality control assessment using FastQC (v0.11.9) to evaluate read quality. Subsequently, adapter trimming and read filtering were performed using Trim-galore (v0.6.5-1) with the following parameters: –phred33 –length 35 –stringency 3. The resulting high-quality reads were aligned to the reference genome (hg38 for humans) using Bowtie2 (v2.2.5) with the parameters: –very-sensitive -x 2000. The uniquely mapped reads were isolated by using samtools view function (samtools v1.6) under parameters: -f 2 -q 10 -b.

## Patient-derived xenografts (PDX) and organoids (PDO)

The HCC patient-derived xenografts (HCC-PDX, No. LI0024, female) were obtained from Shanghai GeneChem Organism by implanting tumor sample fragments (20–30 mm³) derived from an HCC patient into female nude mice (6 weeks old). After one month, twelve mice with tumors reaching an average size of approximately 163 mm³ were selected and randomly divided into 2 groups: six mice in the control group received intraperitoneal injection of 10 μl/g vehicle (10% DMSO, 90% corn oil), and six mice in the test group received SP2509 treatment (i.p., 10 μl/g, 25 mg/kg, twice a week for 3 weeks). Subsequently, tumor xenograft volumes and mice body weights were measured every three days for three weeks. The formula for calculating tumor volume was $V = 0.5 \times a \times b^2$, where a and b represent the long and short diameters of the tumor, respectively. On the 21st day, mice were euthanized. The dissected tumor samples from PDX models were then subjected to the following real-time PCR assays and CUT&Tag-seq assays. The informed consent from all participants was acquired. The use of female mice in the PDX assay is due to the fact that the PDX donor for this experiment was female. Female mice were chosen to maintain experimental consistency and to avoid sex-specific factors that could confound the results. This experimental protocol was approved by the Institutional Animal Care and Use Committee (IACUC) of Shanghai GeneChem Organism and the Ethics Committee of Renji Hospital, Shanghai Jiao Tong University School of Medicine.

HCC patient-derived organoids (HCC-PDO, male) were acquired from D1 Medical Technology (Shanghai, China) through a Material Transfer Agreement (MTA). The organoids were cultured on Matrigel-coated plates to facilitate 3D growth and were maintained in the organoid culture medium (D1 Medical Technology, No. K21103). To evaluate the organoid-forming capacity of the HCC patient-derived cells, organoids were enzymatically dissociated into single cells. Subsequently, these cells were seeded onto Matrigel-coated plates for 3D growth, with a seeding density of 2000 cells per plate. The resulting organoids were then quantified. This study was approved by the Ethics Committee of Renji Hospital, Shanghai Jiao Tong University School of Medicine.

## DepMap database analysis

To investigate the function of histone methylation-related enzymes in hepatocellular carcinoma (HCC) cell viability, we obtained normalized RNAi screening data from the DepMap database (19Q3) via the website: https://depmap.org/portal/download. We selected genes annotated by Gene Ontology (GO) terms GO_HISTONE_DEMETHYLASE_ACTIVITY or GO_HISTONE_LYSINE_N_METHYLTRANSFERASE_ACTIVITY and liver cancer cell lines for analysis. Ultimately, we ranked the mean dependency scores of 62 histone demethylases and methyltransferases in 26 HCC cell lines to evaluate and compare their significance for HCC cell viability. A gene with a negative dependency score is essential for cell growth.

## Motif identification

Motifs significantly enriched in the promoter regions of KDM1A negatively regulated genes were identified using the findMotifs.pl script in the HOMER package (v4.11), with default parameters.

To identify motifs for protein binding or consensus sequences within each family of liver-TEs, BED-formatted files containing genomic regions (MACS2 called peaks, overlapped binding regions, or liver-TE regions) were analyzed using the MEME-ChIP program (v5.4.1), with default parameters. During the MEM-ChIP process, the software MEME and STREME were used to identify novel motifs or consensus sequences. The newly identified motifs were then matched to known motifs in the Cis_BP database or a given set of motifs in MEME format using TOMTOM. The FIMO program was used to screen motifs and find their localizations. Finally, all screened GTF files were merged into

a single BED file for Ngsplot visualizing of CUT&Tag-seq or ATAC-seq intensity profiles surrounding specific motifs.

### LC-MS Metabolite Identification

HepG2 cells treated with SP2509 (n = 2) and vehicle-treated cells (control, n = 2) were subjected to metabolite identification. The untargeted metabolism assays were performed by PANOMIX BioMedical Tech. Co. Suzhou, China. For sample preparation, cells were mixed with glass beads and a mixed solution of acetonitrile, methanol, and water. The mixture was rapidly frozen in liquid nitrogen and thawed, followed by centrifugation to collect the supernatant. The supernatant was then dried and redissolved in an acetonitrile and 2-amino-3-(2-chloro-phenyl)-propionic acid solution. The sample was filtered and transferred for LC-MS analysis.

Liquid Chromatography (LC) analysis was performed on a Vanquish UHPLC System (Thermo Fisher Scientific, USA) with an ACQUITY UPLC ® HSS T3 (150 × 2.1 mm, 1.8 μm) (Waters, Milford, MA, USA) column. Mass spectrometry (MS) was performed on an Orbitrap Exploris 120 instrument (Thermo Fisher Scientific, USA) with an ESI ion source. Full MS-ddMS2 mode with data-dependent MS/MS acquisition was conducted simultaneously. The following parameters were applied: sheath gas pressure of 30 arb, aux gas flow of 10 arb, spray voltage of 3.50 kV for ESI (+) and −2.50 kV for ESI (-), capillary temperature of 325 °C, MS1 range of m/z 100–1000, MS1 resolving power of 60,000 FWHM, 4 data-dependent scans per cycle, MS/MS resolving power of 15,000 FWHM, normalized collision energy of 30%, and automatic dynamic exclusion time.

Metabolite data analysis was performed using MS-DIAL (version 4.8) (http://prime.psc.riken.jp/Metabolomics_Software/MS-DIAL/index.html), an open-source software platform for metabolomic data processing and analysis. The raw LC-MS data files were converted to mzXML format using ProteoWizard MSConvert (version 3.0.22015) and then imported into MS-DIAL for peak alignment and identification. The parameters are as follows: Alignment: Retention time tolerance: 0.05 min; MS' tolerance: 0.015 Da; Identification: Retention time tolerance: 100 min; Accurate mass tolerance (MS1): 0.1 Da; Accurate mass tolerance (MS2): 0.5 Da; Identification score cut off: 50%.

### Statistics & reproducibility

Statistical analyses were conducted by GraphPad Prism 8.0 software or R statistical packages version 4.1.3. Significant differences between groups were examined by Two-sided Student's t-test or Wilcoxon test. P values < 0.05 were considered significant.

No statistical method was used to predetermine the sample size. No data were excluded from the analyses. All mice were randomized before experiments. All experiments, including ChIP-PCR, RT-PCR, western blotting, luciferase reporter assays, immunofluorescence (IF), co-immunoprecipitation (Co-IP), metabolic assays, and in vitro/in vivo functional studies, were independently replicated two or three times with consistent results. The number of replicates performed is detailed in the respective figure legends. Immunohistochemistry (IHC) experiments were conducted on a cohort of 90 HCC tissue samples. The main findings from the high-throughput DNA/RNA sequencing data were validated using other independent sequencing datasets from public repositories. For studies with cell cultures, biological replicates are referred to as independent dishes of cells receiving the same treatment that were processed on the same days or on different days. For each of the following experiments, data collection and analysis were performed blinded to group allocation to ensure unbiased results: MS assays, CUT&Tag-seq, ATAC-seq, and RNA-seq Experiments: Samples were collected in our lab, and the assay procedures were conducted by a third-party company that did not have access to the sample group information. The analyses were performed by a specialist who did not have access to the exact group information until the analysis

was complete. The assays and subsequent data analysis were conducted without knowledge of which samples belonged to which groups. In vitro/ in vivo functional Studies: Cells or animals used in functional assays were conducted by at least two persons. Experimental manipulations and outcome measurements were separately performed without revealing the group identities. IHC Staining: Tissue samples were anonymized. The immunohistochemistry staining and subsequent scoring were performed by researchers blinded to the clinical information and group allocation.

### Reporting summary

Further information on research design is available in the Nature Portfolio Reporting Summary linked to this article.

## Data availability

The sequencing data generated in this study have been deposited in the Gene Expression Omnibus (GEO) repository under the accession codes: GSE228075 (RNA-seq), GSE228072 (RNA-seq) GSE255638 (ATAC-seq), GSE255639 (ATAC-seq), GSE228074 (CUT&Tag-seq), GSE228071 (CUT&Tag-seq), GSE228069 (CUT&Tag-seq), GSE228255 (CUT&Tag-seq), GSE255634 (CUT&Tag-seq), GSE255636 (CUT&Tag-seq), GSE255640 (CUT&Tag-seq), GSE255641 (CUT&Tag-seq), and Sequence Read Archive (SRA) under accession codes: SRP502998 (CUT&Tag-seq), SRP503102 (CUT&Tag-seq and ATAC-seq). The raw data of LC-MS assays are available at integrated proteome resources (iProX) database under the project ID: PXD043284. The raw data of MS-based metabolomics have been deposited in the National Genomics Data Center under the accession code: PRJCA026639. The published data reused in this study includes GSE14520[48] (Expression profile by array), GSE54236[49] (Expression profile by array), and GSM1204463[53] (Bisulfite-Seq) from GEO database, and E-MTAB-7144 (Capture Hi-C) from ArrayExpress database[54]. Several publicly available ChIP-seq raw data were downloaded from the ENCODE database (https://www.encodeproject.org/)[45], including:ENCFF750VZA[52] (ChIP-seq), ENCFF000PJT[52] (ChIP-seq), ENCSR848YWD[52] (ChIP-seq), ENCFF442RQA[52] (ChIP-seq), ENCFF492CBJ[52] (ChIP-seq), ENCFF163SRP[52] (ChIP-seq). The ATAC-seq data of non-tumoral tissues were obtained from BioProject PRJNA63443[45]. The LIHC RNA-seq data from The Cancer Genome Atlas (TCGA) database was derived from the TCGA data portal [https://portal.gdc.cancer.gov/][47]. We downloaded ATAC-seq data (bigWig format) from the NIH GDC data portal [https://gdc.cancer.gov/about-data/publications/ATACseq-AWG][44]. The RNA-seq data of HCC samples from the International Cancer Genome Consortium (ICGC) database[50] was obtained from the ICGC data portal [https://dcc.icgc.org/]. From the UCSC Xena database, we downloaded a combined TCGA and Genotype-Tissue Expression (GTEx) dataset [https://xenabrowser.net/datapages/?cohort=TCGA%20TARGET%20GTEx&removeHub=https%3A%2F%2Fxena.treehouse.gi.ucsc.edu%3A443][51]. The remaining data are available within the Article, Supplementary Information or Source Data file. Source data are provided with this paper.

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

## Acknowledgements

This work was supported by the National Natural Science Foundation of China (82372737 BS.W., 82203441 TT.J., 82273283 YZ.L., 82172905 BS.W., 81972209 BS.W., 82060041 YZ.L., and 82303079 XL.X.) and Shanghai Natural Science Foundation (21ZR1461500 BS.W.).

## Author contributions

TT.J., BS.W., DH.W., YY.J., XL.X., and YZ.L. designed the research, analyzed data, and wrote the manuscript; TT.J., DH.W., XL.X., CS.W., LL.Y., YW.H., YZ.L., YY.J., and BS.W. performed the experiments; TT.J., BS.W., DH.W., and YY.J. revised and edited the paper. YZ.L., YY.J., and BS.W. supervised the project.

## Competing interests

The authors declare no competing interests.
