## [Peer Review File · Nature Communications]

Transposable Elements-Mediated Recruitment of KDM1A
Epigenetically Silences HNF4A Expression to Promote
Hepatocellular CarcinomaReviewers' Comments:

Reviewer #1:

Remarks to the Author:

Jing et al. presented a manuscript that investigates the liver-specific transposable elements mediated mechanisms in hepatocellular carcinoma. They identified that liver TEs suppress HNF4A expression to promote HCC proliferation, and this effect is mediated by the recruitment of lysine demethylase KDM1A to TEs, which epigenetically suppress HNF4A via histone methylation. Further studies demonstrated that besides its transcriptional regulation of HNF4A, KDM1A also interacts with HNF4A complex to suppress its target gene expression, and knockdown or inhibition of KDM1A reversed these effects via upregulation of active histone mark H3K4me1. Importantly, the DNA-binding protein ZMYM3 facilitates the recruitment of KDM1A by liver TEs. In conclusion, this study uncovered the liverTE/KDM1A/HNF4A regulatory axis in the progression of HCC.

Overall, the study is exciting, and extensive transcriptomic and epigenomic analyses support the possible role of TEs/KDM1A in suppressing HNF4A. Although there are several published studies demonstrating the oncogenic function of KDM1A in HCC, this study further investigates its mechanistic effect concerning liver TEs and HNF4A. However, there are significant concerns that reduce the overall enthusiasm for this study which include a lack of 1) robust experimental validation, 2) clinically relevant PDX cell lines, 3) in vivo studies that support the TEs/KDM1A/HNF4A axis in HCC tumor progression, and 4) studies that distinguish KDM1A demethylase dependent/independent functions in regulating HNF4A expression/functions.

1. No rationale was provided for the use of allosteric reversible KDM1A inhibitors (SP2509 and seclidemstat) which inhibit both the catalytic and scaffolding functions of KDM1A as well as reduce KDM1A expression. Authors should determine whether KDM1A-mediated effects on HNF4A expression/target genes are due to KDM1A catalytic or scaffolding activity. This could be done by reintroducing WT and K661A mutant KDM1A in a knockdown/knockout background and by using catalytic irreversible KDM1A inhibitors.
2. It is essential to provide preclinical evidence of the identified regulatory mechanisms on HCC progression. In vivo experiments should be performed to determine the effect of HNF4A-liver-TEs-deletions, KDM1A knockdown with or without HNF4A, and ZMYM3 with or without KDM1A knockdown on HCC tumor growth.
3. Authors should validate key findings using patient-derived HCC cell lines.
4. Fig. 2C. authors observed that KDM1A knockdown increased not only the activation mark H3K4me but also the repressive mark H3K9me2. How do authors explain that? Can they identify genes that are repressed following KDM1A knockdown/inhibition?
5. Fig. 2F, G, H: The oncogenic role of KDM1A is well established in liver cancer, and there are several published reports supporting the oncogenic role of KDM1A. Findings in Fig 2-H are not novel, and this should be moved to supplementary info.
6. Fig. 3A-C: The basal levels of KDM1A in knockout and overexpression conditions seem different, and the KDM1A band is very faint in vector-transfected cells, although the GAPDH levels seem equal in knockdown, inhibitor-treated, and over-expression cells. Does vector alone reduce KDM1A levels in Fig. 3C? If not, please provide KDM1A blots that show relatively comparable levels of basal KDM1A in Fig. 3A-C.
7. Fig. 7. Can the authors perform IHC for ZMYM3 and HNF4A on tissue sections used for KDM1A staining?

8. Fig. 3G,H; 5A-C; 6C, 6G. These findings should be validated in additional HCC cell lines.
9. Colony formation and cell growth assays demonstrated ----- (Fig. 2F). It should be colony formation.
10. Quantification should be provided for Fig. 2F, G. Supp Fig. 2F.
11. Please provide the antibody catalog numbers, and concentrations used for Cut&Tag and western blotting experiments.
12. Page 9, line 3- correct KMD1A to KDM1A

Reviewer #2:

Remarks to the Author:

In the manuscript "Transposable Elements-Mediated Recruitment of KDM1A Epigenetically Silences HNF4A Expression to Promote Hepatocellular Carcinoma" Jing et al. identify TEs that display elevated accessibility in hepatocellular carcinoma (HCC) as well as non-transformed liver cell lines. They demonstrate that these transposable elements contribute to the regulation of numerous genes involved in liver development and differentiation including HNF4A, and that the lysine demethylase KDM1A maintains these TEs in a low-activity or repressed state to promote tumorigenesis. They go on to show that KDM1A physically interacts with many of the transcription factors that act as master regulators of hepatocyte cell fate (e.g. HNF4A, FOXA3, HNF1A), and is also recruited to many TEs via an integration with the transcription factor ZMYM3. At TEs, KDM1A repressed the expression of neighboring genes by reducing the levels of H3K4me1, which in turn has a dramatic effect on the levels of H3K27ac. The authors go on to show that the levels of KDM1A in HCC are prognostic for patient survival, and that the KDM1A and ZMYM3 expression levels in primary HCC samples are anti-correlated with the TE signature of HNF4A target genes. Overall, this manuscript is well written, and provides an interesting novel mechanism for how KDM1A regulates gene expression during hepatocyte development and transformation. This manuscript also helps elucidate the role of TEs in reshaping the genome in the normal and transformed liver. However, I have several major concerns regarding the interpretation of the data that need to be addressed so that I can understand the potential impact of their findings, and whether the conclusions are currently overstated.

Major Concerns:

- (1) In Figure 1F-I the authors show that removing the TEs near HNF4A increases the expression of HNF4A and that driving hyperactivation of these TEs through recruitment of the dCas9-VP64 fusion protein reduces the expression of HNF4A. However, when the authors increase the activity of these TEs by reducing the activity of KDM1A they see an increase in HNF4A expression. This is logically inconsistent. How can increased activity of presumably the same TEs in one case repress the neighboring gene and in the other case activate the neighboring gene?
- (2) The authors show that these same TEs have elevated accessibility in the normal liver as compared to other tissues. Is the recruitment of KDM1A to these TEs in HCC ectopic or does KDM1A regulate these same TEs in the normal liver? Is it possible that by modulating KDM1A they are simply inducing a normal differentiation mechanism rather than suppressing tumor growth per se?
- (3) The results presented in Figure 6E are difficult to interpret. Why in the ZMYM3 knock-down condition is the signal of KDM1A reduced more over all KDM1A peaks than over the ZMYM3 peaks? Can the authors also include the quantification of the KDM1A signal over the KDM1A peaks that overlap with ZMYM3 and those that do not overlap with ZMYM3?

Minor Concerns:

- (1) The authors show that "the enrichment of the histone demethylase KDM1A on the liver-TEs was the most significant" But looking at Figure 2A, there are at least 2-3 genes that are more significantly

enriched than KDM1A. This should be reworded to match the figure.

(2) Figure 2B is not described well in the text.

(3) The authors use the term Transcriptional Regulatory Region (TRR) throughout, is this the same as the TSS? If so, please use TSS. If this is not the same as the TSS, please specifically indicate the region you are referring to on the genome browser tracks.

(4) On multiple genome browser tracks there is a region highlighted in blue. What is this? It is not clearly described in the legend. If this is the TRR please clearly describe this in the legend, and it could be helpful to label it as the TRR on the panel.

(5) Page 9 line 24: "TRR of MAT1A is binded" This should be change to: "TRR of MAT1A is bound"

Reviewer #3:

Remarks to the Author:

Jing et al comprehensively analyses the functional implications of liver-specific transposable elements (TEs) that are activated in hepatocellular carcinoma (HCC) and reveals that these elements play a pivotal role in regulating cellular proliferation through a complex molecular mechanism. The authors uncover a new regulatory axis that involves KDM1A-mediated silencing of HNF4A gene and its targets through the regulatory roles of transposable elements. TE/KDM1A/HNF4A regulatory axis not only offers critical insights into the molecular mechanisms of HCC but also highlights the potential of targeting KDM1A for HCC therapy.

Overall, this study has been carefully designed, the hypothesis and conclusions are based on solid data and very well thought and elegant experiments. This study is of high interest for the TE and cancer fields.

Minor comments:

-For Figure 2C, the authors should provide same profile of histone marks in shuffled TEs (or controls regions) upon KDM1A KD, to rule out any potential sequencing bias. Or snapshot of example regions with control regions should be provided.

-In Fig 3E, the decrease in H3K4me1 is not very clear. Could the authors provide a quantitative comparison with qPCR?

-In Figure 3F, due to unequal loading WBs are hard to interpret for HNF4A expression. The authors should also include RNA level changes of HNF4A upon KDM1A overexpression and deletion of HNF4A-liver-TEs.

-It is unclear whether liver TEs are important regulators of HNF4A gene only or whether they are broad regulators of HNF4A target genes. It will be informative to show the interactions between liver TEs and HNF4A target genes using publicly available Hi-C data for the HCC cell lines.

Reviewer #4:

Remarks to the Author:

In this study, Jing et al. investigated the impact of KDM1A on the regulation of liver-specific transposable elements (TEs) and HNF4A target genes in human hepatocellular carcinoma (HCC). By analyzing publicly available ATAC-Seq data, the authors identified a subset of liver-specific TEs in HCC, including two TE clusters associated with HNF4A. The study demonstrated that KDM1A is enriched in liver-specific TEs and functions to repress the expression of genes associated with these TEs, including HNF4A and MAT1A. Additionally, the authors reported a suppressive interaction between KDM1A and HNF4A in liver cancer. Furthermore, the study suggests that ZMYM3 plays a role in facilitating the

recruitment of KDM1A to liver-specific TEs. While the role of KDM1A in suppressing liver-specific genes and contributing to liver cancer is well-established, the study's exploration of liver-specific TEs, although novel, lacks sufficient evidence and contains several gaps in the proposed hypothesis. Detailed comments are provided below:

1. Mapping transposon elements in the human genome poses challenges due to their significant sequence similarity, which makes differentiation between them difficult. Furthermore, repetitive sequences can cause misalignments and ambiguities during the mapping process. The authors utilized ATAC-Seq data to investigate the chromatin accessibility of TEs in liver cancer; however, they have not adequately addressed how they tackled the alignment issue with TEs.

2. The authors defined liver-specific TEs based on ATAC-Seq data and assigned them to ± 10 kb genes, implying that these TEs likely reside in promoter/enhancer regions of liver-specific genes. Consequently, manipulating these TEs using CRISPR tools directly impacts gene expression. Nevertheless, the authors have not demonstrated whether the accessibility of these TEs differs between HCC and normal liver tissues. Given that the ATAC-Seq data for HCC and normal liver tissues originate from different databases and may lack spike-in controls for cross-sample comparison, the authors should design their own experiments to address this critical question.

3. It is important for the authors to provide specific details regarding the structure and location of the HNF4A-associated TE clusters. Comparing the accessibility of these TE clusters between HCC cell lines and normal liver cells would help to elucidate their role in liver carcinogenesis. The study currently demonstrates that deletion of these TE clusters increases HNF4A expression, while their activation suppresses HNF4A in HCC cell lines. These experiments should be replicated in normal liver cells to establish the significance of these TE clusters in liver cancer development.

4. The authors reported an enrichment of active histone marks, such as H3K4me1, H3K4me2, and H3K27ac, in liver-specific TEs. However, the association between KDM1A and liver-specific TEs appears somewhat unexpected and contradictory to the histone modification enrichment data. This discrepancy arises due to the fact that KDM1A (also known as LSD1) functions as a histone demethylase of H3K4me. Therefore, the authors should provide an explanation for this contradiction. Furthermore, the authors observed an increase in H3K9me2 (a repressive histone mark) in liver-specific TEs after KDM1A knockdown (Figure 2C). However, these findings do not entirely support the authors' conclusion that KDM1A down-regulation enhances the activation and accessibility of liver-specific TEs and nearby chromatin. To strengthen their argument, the authors should perform ATAC-Seq experiments.

5. Although knockdown of KDM1A or treatment with a KDM1A inhibitor suppressed HCC colony formation and tumor growth in nude mice, it is challenging to attribute these phenotypic changes solely to alterations in liver-specific TEs.

6. The authors observed only marginal effects of KDM1A knockdown on the histone modification profile of the HNF4A promoter (or its associated liver-specific TE clusters). ATAC-Seq data is also missing, which would help to clarify the extent of KDM1A's impact on the accessibility of liver-specific TEs.

7. The authors claimed that KDM1A interacts with HNF4A to suppress the expression of HNF4A target genes. This conclusion appears counterintuitive to the common understanding of HNF4A as a master transcription factor that drives the expression of liver-specific genes and maintains liver cell identity. Therefore, the authors should provide additional evidence to support this claim.

8. The authors presented data suggesting that ZMYM3 plays a role in facilitating KDM1A DNA binding and demethylation in HNF4A target genes. However, the authors did not provide a mechanistic explanation for how ZMYM3 mediates the binding of KDM1A to liver-specific TEs. Therefore, the

authors should provide more experimental evidence to support this mechanism.

9. The reviewer raised the question of whether there is any overlap between liver-specific TE-associated genes and HNF4A target genes. Investigating this overlap would help to elucidate the role of liver-specific TEs in HCC and whether they contribute to the suppression of HNF4A target genes.

Reviewer #1, expertise in KDM1A (Remarks to the Author):

Jing et al. presented a manuscript that investigates the liver-specific transposable elements mediated mechanisms in hepatocellular carcinoma. They identified that liver TEs suppress HNF4A expression to promote HCC proliferation, and this effect is mediated by the recruitment of lysine demethylase KDM1A to TEs, which epigenetically suppress HNF4A via histone methylation. Further studies demonstrated that besides its transcriptional regulation of HNF4A, KDM1A also interacts with HNF4A complex to suppress its target gene expression, and knockdown or inhibition of KDM1A reversed these effects via upregulation of active histone mark H3K4me1. Importantly, the DNA-binding protein ZMYM3 facilitates the recruitment of KDM1A by liver TEs. In conclusion, this study uncovered the liverTE/KDM1A/HNF4A regulatory axis in the progression of HCC.

Overall, the study is exciting, and extensive transcriptomic and epigenomic analyses support the possible role of TEs/KDM1A in suppressing HNF4A. Although there are several published studies demonstrating the oncogenic function of KDM1A in HCC, this study further investigates its mechanistic effect concerning liver TEs and HNF4A. However, there are significant concerns that reduce the overall enthusiasm for this study which include a lack of 1) robust experimental validation, 2) clinically relevant PDX cell lines, 3) in vivo studies that support the TEs/KDM1A/HNF4A axis in HCC tumor progression, and 4) studies that distinguish KDM1A demethylase dependent/independent functions in regulating HNF4A expression/functions.

1. No rationale was provided for the use of allosteric reversible KDM1A inhibitors (SP2509 and seclidemstat) which inhibit both the catalytic and scaffolding functions of KDM1A as well as reduce KDM1A expression. Authors should determine whether KDM1A-mediated effects on HNF4A expression/target genes are due to KDM1A catalytic or scaffolding activity. This could be done by reintroducing WT and K661A mutant KDM1A in a knockdown/knockout background and by using catalytic irreversible KDM1A inhibitors.

Response:

In response to the reviewer's constructive suggestion, we conducted supplementary experiments to elucidate whether KDM1A-mediated regulation of HNF4A expression relies on its catalytic activity. We engineered Flag-tagged and RNAi-resistant expression vectors for wild-type KDM1A and the K661A mutant variant. These constructs were subsequently reintroduced into KDM1A knockdown cells. Our findings reveal that while wild-type KDM1A effectively mitigated the KDM1A-knockdown-induced elevation of HNF4A, the K661A mutant failed to elicit such a response (**Fig. S3e**). This observation suggests that the catalytic activity of KDM1A is crucial for its regulation of HNF4A expression.

Furthermore, as per the reviewer's recommendation, we investigated the effects of several irreversible KDM1A inhibitors on HCC cells. Our results demonstrate that these irreversible inhibitors elicited comparable effects to the reversible inhibitors SP2509 and seclidemstat, evidenced by the augmentation of H3K4me1 modification (**Fig. S2g**) in liver-TEs and the upregulation of HNF4A expression (**Fig. S3f**).

2. It is essential to provide preclinical evidence of the identified regulatory mechanisms on HCC progression. In vivo experiments should be performed to determine the effect of HNF4A-liver-TEs-deletions, KDM1A knockdown with or without HNF4A, and ZMYM3 with or without KDM1A knockdown on HCC tumor growth.

Response:

In accordance with your valuable suggestion, we conducted the recommended *in vivo* experiments to investigate the regulatory mechanisms identified in our study on hepatocellular carcinoma progression. Our results revealed that the deletion of HNF4A-liver-TEs induced a significant inhibition of HCC cell growth in nude mice (**Fig. S1j**). Moreover, the knockdown of KDM1A markedly suppressed tumor cell growth *in vivo*.

However, this effect was mitigated by the concurrent downregulation of HNF4A, underscoring the essential role of HNF4A in the dysregulation mediated by KDM1A (**Fig. S3l**). Furthermore, the knockdown of ZMYM3 resulted in a complete elimination of the *in vivo* tumorigenic potential of HCC cells (**Fig. S6a**), underscoring the critical role of ZMYM3 in HCC tumorigenesis. These findings provide compelling preclinical evidence supporting the regulatory roles of HNF4A-liver-TEs, KDM1A/HNF4A, and ZMYM3 in HCC progression.

3. Authors should validate key findings using patient-derived HCC cell lines.

Response:

We appreciate the insightful recommendation provided by the reviewer regarding the validation of our key findings using patient-derived materials. To address this, we employed two HCC patient-derived models: patient-derived xenografts (PDX) and patient-derived organoids (PDO).

In our investigation using the PDX model, we conducted experiments to assess the *in vivo* inhibitory effects of the KDM1A inhibitor. Our results demonstrated a significant retardation of tumor growth upon SP2509 treatment, while no alterations in body weights were observed (**Fig. 2h**). Subsequently, we analyzed tumor samples from these experiments and observed an increase in the H3K4me1 within liver-TEs (**Fig. 2i**) and an upregulation of HNF4A expression (**Fig. S3c**) following KDM1A inhibition. Concurrently, we cultured PDX samples *in vitro* to establish a patient-derived cell line. Depletion of HNF4A-liver-TEs and KDM1A knockdown exerted notable suppressive effects on tumor cell growth (**Fig. S2l**), consistent with the observed effects in conventional HCC cell lines (**Fig. 1h and Fig. 2g**). Moreover, by utilizing the PDO model, we investigated the impact of HNF4A-liver-TEs depletion and KDM1A inhibition on the 3D growth of HCC cells. Our findings revealed significant inhibition of 3D growth upon depletion of HNF4A-liver-TEs and inhibition of KDM1A (**Fig. S1k and Fig. 2j**).

4. Fig. 2C. authors observed that KDM1A knockdown increased not only the activation mark H3K4me but also the repressive mark H3K9me2. How do authors explain that? Can they identify genes that are repressed following KDM1A knockdown/inhibition?

Thank you for your insightful comment. In response to your concerns regarding the observed increase in both the activation mark H3K4me1 and the repressive mark H3K9me2 upon KDM1A knockdown, we have formulated a hypothesis suggesting that KDM1A may regulate histone marks through either a direct or indirect mechanism. To discern the direct regulation of specific histone marks by KDM1A, we conducted a detailed examination of how KDM1A knockdown influences H3K4me1 and H3K9me2 marks in the transcriptional regulatory regions bound by KDM1A. Our results demonstrate that in regions directly targeted by KDM1A, H3K4me1 exhibited up-regulation following KDM1A knockdown, while H3K9me2 showed no significant change (**Fig. S2e, Top panel**). This observation is consistently supported by findings in Liver-TE-associated KDM1A-targeted regions, where H3K4me1 marks also significantly increased upon KDM1A knockdown (**Fig. S2e, Down panel**). These results strongly indicate that KDM1A may directly regulate H3K4me1, leading us to focus our subsequent analyses on this mark.

Given that H3K4me1 is negatively regulated by KDM1A and associated with transcriptional activation, we selected KDM1A-negatively regulated genes—those up-regulated following KDM1A down-regulation—for further investigation. However, we recognize the importance of KDM1A-positively regulated genes and have included a comprehensive list of these genes in the **Supplementary Table 6** of the revised manuscript.

5. Fig. 2F, G, H: The oncogenic role of KDM1A is well established in liver cancer, and there are several published reports supporting the oncogenic role of KDM1A. Findings in Fig 2-H are not novel, and this should be moved to supplementary info.

Response:

We highly agree with this suggestion. In response, we have relocated this result to the supplementary figures.

6. Fig. 3A-C: The basal levels of KDM1A in knockout and overexpression conditions seem different, and the KDM1A band is very faint in vector-transfected cells, although the GAPDH levels seem equal in knockdown, inhibitor-treated, and overexpression cells. Does vector alone reduce KDM1A levels in Fig. 3C? If not, please provide KDM1A blots that show relatively comparable levels of basal KDM1A in Fig. 3A-C.

Response:

The vector does not reduce KDM1A levels. The observed difference is due to different exposure levels. We have provided the blots showing comparable levels of basal KDM1A in the revised figure.

7. Fig. 7. Can the authors perform IHC for ZMYM3 and HNF4A on tissue sections used for KDM1A staining?

Response:

Yes, we have conducted immunohistochemical staining using antibodies targeting ZMYM3 and HNF4A on the same set of tissue samples used for KDM1A staining. Subsequent correlation analyses revealed a negative association between both KDM1A and ZMYM3 with HNF4A expression in HCC samples. These results were provided in the **Fig. S7a-b**.

8. Fig. 3G,H; 5A-C; 6C, 6G. These findings should be validated in additional HCC cell lines.

Response:

In response to your suggestion, we have validated all these findings in at least one additional HCC cell line. The results are depicted in Fig. 3g, h; Fig. 5a-c; Fig. 6c, g; Fig. S6d.

9. Colony formation and cell growth assays demonstrated ----- (Fig. 2F). It should be colony formation.

In response:

As per your suggestion, we have corrected this mistake.

10. Quantification should be provided for Fig. 2F, G. Supp Fig. 2F.

Response:

In the revised manuscript, we have included the quantification data and statistical analyses for these experiments.

11. Please provide the antibody catalog numbers, and concentrations used for Cut&Tag and western blotting experiments.

Response:

In the revised Materials and Methods section, we have incorporated catalog numbers, and concentrations of the antibodies used in the study (Page 29, line 828-831).

12. Page 9, line 3- correct KMD1A to KDM1A

We have corrected this mistake.

Reviewer #2, expertise in Cut&Tag (Remarks to the Author):

In the manuscript “Transposable Elements-Mediated Recruitment of KDM1A Epigenetically Silences HNF4A Expression to Promote Hepatocellular Carcinoma” Jing et al. identify TEs that display elevated accessibility in hepatocellular carcinoma (HCC) as well as non-transformed liver cell lines. They demonstrate that these transposable elements contribute to the regulation of numerous genes involved in liver development and differentiation including HNF4A, and that the lysine demethylase KDM1A maintains these TEs in a low-activity or repressed state to promote tumorigenesis. They go on to show that KDM1A physically interacts with many of the transcription factors that act as master regulators of hepatocyte cell fate (e.g. HNF4A, FOXA3, HNF1A), and is also recruited to many TEs via an integration with the transcription factor ZMYM3. At TEs, KDM1A repressed the expression of neighboring genes by reducing the levels of H3K4me1, which in turn has a dramatic effect on the levels of H3K27ac. The authors go on to show that the levels of KDM1A in HCC are prognostic for patient survival, and that the KDM1A and ZMYM3 expression levels in primary HCC samples are anti-correlated with the TE signature of HNF4A target genes. Overall, this manuscript is well written, and provides an interesting novel mechanism for how KDM1A regulates gene expression during hepatocyte development and transformation. This manuscript also helps elucidate the role of TEs in reshaping the genome in the normal and transformed liver. However, I have several major concerns regarding the interpretation of the data that need to be addressed so that I can understand the potential impact of their findings, and whether the conclusions are currently overstated.

Major Concerns:

(1) In Figure 1F-I the authors show that removing the TEs near HNF4A increases the expression of HNF4A and that driving hyperactivation of these TEs through recruitment of the dCas9-VP64 fusion protein reduces the expression of HNF4A. However, when the authors increase the activity of these TEs by reducing the activity of KDM1A they see an increase in HNF4A expression. This is logically

inconsistent. How can increased activity of presumably the same TEs in one case repress the neighboring gene and in the other case activate the neighboring gene?

Response:

We appreciate this invaluable feedback. To specifically address this concern, we conducted the KDM1A CUT&Tag-seq assay subsequent to recruiting dCas9-VP64 through sgRNA to target HNF4A-liver-TEs. Our findings demonstrated a noteworthy facilitation of KDM1A binding towards HNF4A-liver-TEs induced by dCas9-VP64 (**Fig. S3i**). This observation aligns with previous findings indicating that the deletion of HNF4A-liver-TEs reduced KDM1A binding (**Fig. 3g**), suggesting opposing roles for dCas9-VP64 and TE deletion in KDM1A recruitment.

Given our comprehensive demonstration that KDM1A suppresses HNF4A expression (**Fig. 3a-c**), it is reasonable for us to propose that HNF4A-liver-TEs regulate HNF4A expression through the recruitment of KDM1A. In our pursuit to elucidate the pivotal role of KDM1A in regulating liver-TEs-mediated HNF4A expression, we employed a knockdown approach against KDM1A expression in HCC cells expressing dCas9-VP64/HNF4A-liver-TE-sgRNA. As anticipated, the inhibition of KDM1A expression resulted in a profound restoration of HNF4A expression levels that were previously diminished by dCas9-VP64 (**Fig. S3j**). This restoration of HNF4A expression underscores a direct regulatory influence exerted by KDM1A on HNF4A expression modulated by liver-TEs.

Collectively, our supplementary experiments provide insights into a crucial mechanism: the accessibility of HNF4A-liver-TE regions directly correlates with the recruitment of KDM1A, which, in turn, inhibits HNF4A, thereby assigning HNF4A-liver-TEs a negative regulatory role in governing HNF4A expression. This mechanistic understanding effectively rationalizes the observed effects of dCas9-VP64 and HNF4A-liver-TE-deletion, both of which rely on the recruitment of KDM1A. Specifically, dCas9-VP64 facilitates the recruitment of KDM1A, leading to the attenuation of HNF4A expression. Conversely, the deletion of HNF4A-liver-TEs results in the dissociation of KDM1A, thereby promoting the expression of HNF4A.

(2) The authors show that these same TEs have elevated accessibility in the normal liver as compared to other tissues. Is the recruitment of KDM1A to these TEs in HCC ectopic or does KDM1A regulate these same TEs in the normal liver? Is it possible that by modulating KDM1A they are simply inducing a normal differentiation mechanism rather than suppressing tumor growth per se?

Response:

In response to this insightful inquiry, we conducted KDM1A CUT&Tag-seq assays in both normal liver cells (THLE2, THLE3) and HCC cell lines to compare the binding ability of KDM1A within liver-specific TEs. Our findings revealed a weaker binding ability of KDM1A to these TEs in normal liver cells compared to HCC cells (**Fig. S2c**). These results suggest a potential aberrant recruitment of KDM1A to liver-specific TEs within the context of liver cancer. Moreover, consistent with these observations, the inhibition of KDM1A in normal liver cells failed to elevate H3K4me1 modification of liver-TEs (**Fig. S2f**) or increase HNF4A expression (**Fig. S3h**), contrasting with the effects observed in HCC cells. These findings strongly indicate that the impact of KDM1A on HNF4A is distinctly specific to liver cancer and is not associated with normal liver development.

(3) The results presented in Figure 6E are difficult to interpret. Why in the ZMYM3 knock-down condition is the signal of KDM1A reduced more over all KDM1A peaks than over the ZMYM3 peaks? Can the authors also include the quantification of the KDM1A signal over the KDM1A peaks that overlap with ZMYM3 and those that do not overlap with ZMYM3?

Response:

We deeply value this insightful feedback provided by the reviewer. To address this concern, we applied a quantification and filtration process to the KDM1A peaks based on MASC2 scores and length criteria (scores >20 and length >500bp) for analysis.

Furthermore, recognizing the potential bias associated with directly using ZMYM3 peaks from the public ENCODE database, we utilized the MEME suite to predict and subsequently search for ZMYM3 motifs within the human genome. Subsequently, in our revised manuscript, we conducted the comparative analysis of the KDM1A binding signal surrounding these identified ZMYM3 motifs.

Our results, depicted in the revised Fig. 6e, indicated a consistent reduction in the KDM1A binding signal around both KDM1A peaks and ZMYM3 motifs. Additionally, in response to the reviewer's request, when assessing the KDM1A binding ability in regions with or without ZMYM3 motifs contained by KDM1A peaks, we observed a significantly heightened binding capacity of KDM1A in regions harboring ZMYM3 motifs (**Fig. S6g**). This observation strongly implies a potential role of ZMYM3 in enhancing the DNA binding capability of KDM1A.

Minor Concerns:

(1) The authors show that “the enrichment of the histone demethylase KDM1A on the liver-TEs was the most significant” But looking at Figure 2A, there are at least 2-3 genes that are more significantly enriched than KDM1A. This should be reworded to match the figure.

Response:

We are grateful to the reviewer for the careful observation, bringing attention to this discrepancy. We have revised the statement to better align with the data.

(2) Figure 2B is not described well in the text.

Response:

We have incorporated a more detailed description of the findings presented in Fig. 2b, specifically highlighting the significant enrichment of KDM1A signals within the liver-TEs compared to similarly sized shuffled TE regions (Page 7, line 177-181).

(3) The authors use the term Transcriptional Regulatory Region (TRR) throughout, is this the same as the TSS? If so, please use TSS. If this is not the same as the TSS, please specifically indicate the region you are referring to on the genome browser tracks.

Response:

The term "Transcriptional Regulatory Region" (TRR) in our manuscript denotes the genomic region within ± 10 kb around the Transcription Start Site (TSS). We have now explicitly indicated this region on the genome browser tracks (**Fig. 3e and Fig. 5e**).

(4) On multiple genome browser tracks there is a region highlighted in blue. What is this? It is not clearly described in the legend. If this is the TRR please clearly describe this in the legend, and it could be helpful to label it as the TRR on the panel.

Response:

The regions initially highlighted in blue on the genome browser tracks denoted altered histone mark peaks. In response to your valuable suggestion and to prevent any potential confusion, we have removed these labels. Furthermore, to enhance clarity, we have now explicitly labeled the Transcriptional Regulatory Region (TRR) on the panels for better interpretation.

(5) Page 9 line 24: "TRR of MAT1A is binded" This should be change to: "TRR of MAT1A is bound"

Response:

We have corrected this mistake accordingly.

Reviewer #3, expertise in transposable elements in cancer (Remarks to the Author):

Jing et al comprehensively analyses the functional implications of liver-specific transposable elements (TEs) that are activated in hepatocellular carcinoma (HCC) and reveals that these elements play a pivotal role in regulating cellular proliferation through a complex molecular mechanism. The authors uncover a new regulatory axis that involves KDM1A-mediated silencing of HNF4A gene and its targets through the regulatory roles of transposable elements. TE/KDM1A/HNF4A regulatory axis not only offers critical insights into the molecular mechanisms of HCC but also highlights the potential of targeting KDM1A for HCC therapy.

Overall, this study has been carefully designed, the hypothesis and conclusions are based on solid data and very well thought and elegant experiments. This study is of high interest for the TE and cancer fields.

Minor comments:

-For Figure 2C, the authors should provide same profile of histone marks in shuffled TEs (or controls regions) upon KDM1A KD, to rule out any potential sequencing bias. Or snapshot of example regions with control regions should be provided.

Response:

We appreciate the insightful suggestion raised by the reviewer regarding Figure 2C. In response to this constructive feedback, we have incorporated the profile of histone marks in shuffled TEs in the revised Figure 2. Furthermore, we specifically selected a TE situated within the transcriptional regulatory region of AFP, a liver-TE-associated gene, as example region (see the graph below).

-In Fig 3E, the decrease in H3K4me1 is not very clear. Could the authors provide a quantitative comparison with qPCR?

Response:

In response to this concern, we have conducted ChIP-RT-PCR assays specifically focusing on the transcriptional regulatory region of the HNF4A gene. The newly generated data is presented in **Fig. S3d**, allowing for a quantitative comparison of H3K4me1 modification levels.

-In Figure 3F, due to unequal loading WBs are hard to interpret for HNF4A expression. The authors should also include RNA level changes of HNF4A upon KDM1A overexpression and deletion of HNF4A-liver-TEs.

Response:

To address this concern, we repeated these experiments and obtained results consistent with the previous study. Additionally, we validated these findings in an additional HCC cell line. Furthermore, we have included real-time PCR assays in the revised manuscript to demonstrate the expression changes of HNF4A (**Fig. 3f**).

-It is unclear whether liver TEs are important regulators of HNF4A gene only or whether they are broad regulators of HNF4A target genes. It will be informative to show the interactions between liver TEs and HNF4A target genes using publicly available Hi-C data for the HCC cell lines.

Response:

In response to your insightful suggestion, we conducted an analysis using a Capture HiC data obtained from the ArrayExpress database (E-MTAB-7144), which was derived from HepG2 cells. Our findings indicate that genomic interactions involving HNF4A-liver-TEs primarily constitute cis-interactions, predominantly localized in the q13.12 region of chromosome 20.

Upon closer examination, we observed that the genes interacting with these elements do not include robust HNF4A target genes. Despite considering the potential long-range genomic interactions, our analysis suggests that HNF4A-liver-TEs do not function as broad regulators of HNF4A target genes. These observations have been incorporated into the revised **Fig. S1h** for clarity.

Reviewer #4, expertise in liver cancer epigenetics (Remarks to the Author):

In this study, Jing et al. investigated the impact of KDM1A on the regulation of liver-specific transposable elements (TEs) and HNF4A target genes in human hepatocellular carcinoma (HCC). By analyzing publicly available ATAC-Seq data, the authors identified a subset of liver-specific TEs in HCC, including two TE clusters associated with HNF4A. The study demonstrated that KDM1A is enriched in liver-specific TEs and functions to repress the expression of genes associated with these TEs, including HNF4A and MAT1A. Additionally, the authors reported a suppressive interaction between KDM1A and HNF4A in liver cancer. Furthermore, the study suggests that ZMYM3 plays a role in facilitating the recruitment of KDM1A to liver-specific TEs. While the role of KDM1A in suppressing liver-specific genes and contributing to liver cancer is well-established, the study's exploration of liver-specific TEs, although novel, lacks sufficient evidence and contains several gaps in the proposed hypothesis. Detailed comments are provided below:

1. Mapping transposon elements in the human genome poses challenges due to their significant sequence similarity, which makes differentiation between them difficult. Furthermore, repetitive sequences can cause misalignments and ambiguities during the mapping process. The authors utilized ATAC-Seq data to investigate the chromatin accessibility of TEs in liver cancer; however, they have not adequately addressed how they tackled the alignment issue with TEs.

Response:

Mapping transposon elements (TEs) indeed presents a significant challenge due to their considerable sequence similarity, leading to difficulties in distinguishing between them. Moreover, repetitive sequences contribute to misalignments and uncertainties during the mapping process. We appreciate your scrutiny regarding the alignment issue,

specifically in utilizing ATAC-Seq data to explore the chromatin accessibility of TEs in liver cancer.

Addressing the alignment issue when investigating transposon elements (TEs) in the human genome is a pivotal challenge encountered not only in our study but also in various TE-related investigations. However, several methodologies have been established to address this concern, including the exclusion of multiple-mapped and low-quality reads while isolating uniquely mapped, high-quality reads subsequent to the alignment process (Corces et al., *Science*. 2018 Oct 26;362(6413): eaav1898; Pal et al., *Nat Struct Mol Biol*. 2023 Jul;30(7):935-947; Jumpei et al., *PLoS Genet*. 2017 Jul 12;13(7):e1006883).

In our study, the ATAC-seq data utilized was sourced from the publicly available TCGA database, originally generated by Corces et al. (*Science*. 2018 Oct 26;362(6413): eaav1898). Notably, the methodology employed in the original study preemptively addressed mappability concerns by preserving uniquely mapped reads after alignment using samtools software with the following parameter: `samtools view -f 2 -q 10 -b`. Consequently, the obtained transformed read coverage files (in bigwig format) from Corces's work underwent processing aimed at minimizing the likelihood of misalignments and ambiguities. It's worth noting that the ATAC-seq data from Corces's work has been successfully utilized for TE-related data mining in another significant study (Jumpei et al., *Sci Adv*. 2020 Oct 21;6(43):eabc3020), indicating the validity of our approach.

In our revised manuscript, we introduced further strategies to address alignment challenges arising from the repetitive nature of TEs. For the TCGA ATAC-seq dataset, beyond exploring the chromatin accessibility of TE loci, we broadened our investigation to the transcriptional regulatory regions (TSS \pm 10kb) that encompass liver-specific accessible TEs, referred to as liver-TE-TRRs. These regions denote unique genomic areas with mappability. As anticipated, the liver-TE-TRRs exhibited notably high accessibility (**Fig. S1a**), aligning consistently with the accessibility pattern observed for liver-TEs.

Regarding the ATAC-seq data sourced from the ENCODE database, we retrieved the BAM formatted files and conducted filtration of uniquely mapped reads using samtools

with the previously mentioned parameters. Subsequently, BAM files containing highly qualified alignment reads were subjected to further analysis. Our findings from this dataset also corroborate the liver-specific accessibility of liver-TEs (**Fig. 1c**).

2. The authors defined liver-specific TEs based on ATAC-Seq data and assigned them to ± 10 kb genes, implying that these TEs likely reside in promoter/enhancer regions of liver-specific genes. Consequently, manipulating these TEs using CRISPR tools directly impacts gene expression. Nevertheless, the authors have not demonstrated whether the accessibility of these TEs differs between HCC and normal liver tissues. Given that the ATAC-Seq data for HCC and normal liver tissues originate from different databases and may lack spike-in controls for cross-sample comparison, the authors should design their own experiments to address this critical question.

Response:

We highly appreciate your insightful comment regarding the accessibility of liver-TEs in HCC versus normal liver tissues. In response to this concern, we conducted additional experiments to compare the accessibility of these liver-TEs in healthy liver cells and cancerous liver cells.

We utilized two normal liver cell lines (THLE2, THLE3) and two liver cancer cell lines (Huh7, HepG2) to investigate liver-TE accessibility through ATAC-seq experiments. Following the isolation of uniquely mapped reads, we discerned that liver-TEs exhibited greater accessibility in normal liver cells (**Fig. S1b, left panel**). In concurrence, our analysis revealed heightened accessibility of liver-TE-associated transcriptional regulatory regions (liver-TE-TRRs) in normal liver cells (**Fig. S1b, right panel**). These findings suggest a higher likelihood of accessibility for liver-TEs in normal liver cells.

3. It is important for the authors to provide specific details regarding the structure and location of the HNF4A-associated TE clusters. Comparing the accessibility of

these TE clusters between HCC cell lines and normal liver cells would help to elucidate their role in liver carcinogenesis. The study currently demonstrates that deletion of these TE clusters increases HNF4A expression, while their activation suppresses HNF4A in HCC cell lines. These experiments should be replicated in normal liver cells to establish the significance of these TE clusters in liver cancer development.

Response:

In response to this valuable suggestion, we have included an Integrative Genomics Viewer (IGV) snapshot (**Fig. S1f**) and a schematic diagram (**Fig. S1i**) illustrating the genomic locations of the HNF4A-associated TE clusters.

Furthermore, we conducted ATAC-seq assays in two normal liver cell lines (THLE2 and THLE3) and two HCC cell lines (Huh7 and HepG2). The comparison of ATAC-seq signal intensity specifically focused on HNF4A-liver-TEs, revealing distinct accessibility patterns. Notably, these TE clusters exhibited greater accessibility in normal liver cells compared to tumoral cells, as depicted in **Fig. S1g**.

Regarding your subsequent inquiry, we performed supplementary experiments to explore the regulatory role of HNF4A-liver-TEs in normal liver cells. The results indicate that, unlike in HCC cells, HNF4A-liver-TEs did not significantly impact HNF4A expression in normal liver cells (**Fig. S3g**). Furthermore, our investigation unveiled a notable distinction in the binding pattern of KDM1A to these TEs in normal and tumoral liver cells. Specifically, the binding of KDM1A to HNF4A-liver-TEs was substantially weakened in normal liver cells compared to HCC cells (**Fig. S2c**). Given that the recruitment of KDM1A is crucial for the inhibitory role of these TEs on HNF4A, the diminished binding in normal liver cells may explain the observed lack of regulatory effect.

4. The authors reported an enrichment of active histone marks, such as H3K4me1, H3K4me2, and H3K27ac, in liver-specific TEs. However, the association between KDM1A and liver-specific TEs appears somewhat unexpected and contradictory to the histone modification enrichment data. This discrepancy arises due to the fact that KDM1A (also known as LSD1) functions as a histone demethylase of H3K4me. Therefore, the authors should provide an explanation for this contradiction. Furthermore, the authors observed an increase in H3K9me2 (a repressive histone mark) in liver-specific TEs after KDM1A knockdown (Figure 2C). However, these findings do not entirely support the authors' conclusion that KDM1A down-regulation enhances the activation and accessibility of liver-specific TEs and nearby chromatin. To strengthen their argument, the authors should perform ATAC-Seq experiments.

Response:

Our findings indicate the enrichment of both the histone marker H3K4me1 and its demethylase KDM1A at the liver-TEs, suggesting a complex regulation of chromatin states in these regions. The co-enrichment of both H3K4me1 and KDM1A at the liver-TEs may seem contradictory, but we propose several possible explanations for this observation.

One possible explanation is that KDM1A modulates H3K4me1 levels at the liver-TE sites to fine-tune their transcriptional regulatory activity. KDM1A has been shown to function as a transcriptional co-repressor and may act in concert with other chromatin regulators to silence gene expression. In this context, the presence of KDM1A at liver-TE could help to prevent aberrant activation of nearby genes by modulating the levels of H3K4me1. Another possibility is that liver-TE represents a transitional state between active and inactive enhancers, where H3K4me1 and KDM1A coexist. In this scenario, the balance between H3K4me1 and KDM1A levels may determine whether the TE-derived cis-regulatory elements (CRE) is active or repressed. Low levels of KDM1A may allow for H3K4me1 to persist and maintain an active state around TE-derived CRE, whereas high levels of KDM1A may result in H3K4me1 removal and subsequent silencing. Overall, the co-enrichment of H3K4me1 and KDM1A at the liver-TEs suggests a

complex regulation of transactivation activity in this region, which is consistent with the high expression of KDM1A in liver cancer and the low expression of liver-TE related genes in liver cancer compared to normal liver tissue. In the revised manuscript, we have discussed this important point in the discussion section (Page 15, Line 417-Page 16, Line 439).

Regarding the observed increase in both the activation mark H3K4me and the repressive mark H3K9me2 upon KDM1A knockdown, we have proposed a hypothesis suggesting that KDM1A may regulate histone markers through either a direct or indirect mechanism. To discern the direct regulation of specific histone marks by KDM1A, we conducted a detailed examination of how KDM1A knockdown influences H3K4me1 and H3K9me2 marks in the transcriptional regulatory regions bound by KDM1A. Our results demonstrate that in regions directly targeted by KDM1A, H3K4me1 exhibited up-regulation following KDM1A knockdown, while H3K9me2 showed no significant change (**Fig. S2e, Top panel**). This observation is consistently supported by findings in Liver-TE-associated KDM1A-targeted regions, where H3K4me1 marks also significantly increased upon KDM1A knockdown (**Fig. S2e Down panel**). These results strongly indicate that KDM1A may directly regulate H3K4me1.

As suggested by the reviewer, we have performed the ATAC-seq assays, which revealed a significant elevation of accessibility of liver-TEs upon KDM1A knockdown (**Fig. 2c**). This observation is inconsistent with the enrichment of H3K4me1 modification after KDM1A down-regulation.

5. Although knockdown of KDM1A or treatment with a KDM1A inhibitor suppressed HCC colony formation and tumor growth in nude mice, it is challenging to attribute these phenotypic changes solely to alterations in liver-specific TEs.

Response:

Yes, indeed, KDM1A is a multifunctional protein, and its regulatory role has been identified in various types of tumors, including liver cancer, through different molecular mechanisms. In our study, we observed that Liver-TEs, especially HNF4A-liver-TEs,

play a crucial role in recruiting KDM1A, representing one of the key mechanisms by which it influences liver cancer cell growth as cis-elements. The biological function of HNF4A-liver-TEs is dependent on KDM1A. This is evident from our observation that the loss of HNF4A-liver-TE inhibits the binding of KDM1A to the transcriptional regulatory region of the HNF4A gene (**Fig. 3g**), subsequently affecting the expression of the HNF4A gene (**Fig. 3f**) and influencing cell growth (**Fig. 3h**).

In our study, we also unveiled a novel mechanism by which KDM1A affects tumor cell growth. KDM1A inhibits HNF4A expression, promoting liver cancer cell growth. To demonstrate the contributory role of HNF4A in KDM1A-induced oncogenic effects, we conducted additional experiments. We found that inhibiting HNF4A expression in KDM1A-knockdown HCC cells restored their growth (**Fig. S3k**). Additionally, through *in vivo* experiments, we demonstrated that inhibiting HNF4A expression could counteract the attenuating effect on HCC cell tumorigenicity induced by KDM1A downregulation (**Fig. S3l**). In summary, through supplementary experiments, we have validated that HNF4A is a critical downstream effector responsible for the biological phenotypes induced by KDM1A.

6. The authors observed only marginal effects of KDM1A knockdown on the histone modification profile of the HNF4A promoter (or its associated liver-specific TE clusters). ATAC-Seq data is also missing, which would help to clarify the extent of KDM1A's impact on the accessibility of liver-specific TEs.

Response:

In response to your suggestion, we conducted ATAC-seq assays following KDM1A knockdown. We have included the ATAC-seq signal intensity changes surrounding liver-TEs (**Fig. 2c**). Specifically, we have presented the accessibility alterations in the HNF4A-liver-TEs and the adjacent regions using the IGV browser (**Fig. 3e**).

7. The authors claimed that KDM1A interacts with HNF4A to suppress the expression of HNF4A target genes. This conclusion appears counterintuitive to the common understanding of HNF4A as a master transcription factor that drives the expression of liver-specific genes and maintains liver cell identity. Therefore, the authors should provide additional evidence to support this claim.

Response:

In response to this concern, we conducted additional experiments wherein exogenous HNF4A was overexpressed in HCC cells, leading to a significant increase in the expression of genes associated with liver cell-specific functions, such as MAT1A (**Fig. S5a**). This approach phenocopied the effects observed in KDM1A knockdown, particularly on the expression of these genes. Intriguingly, the down-regulation of HNF4A in KDM1A-knockdown cells counteracted the elevation of these genes induced by KDM1A knockdown (**Fig. S5b**). These findings suggest that HNF4A functions as a master transcription factor driving the expression of liver-specific genes, and the suppression of HNF4A is essential for the oncogenic role of KDM1A.

8. The authors presented data suggesting that ZMYM3 plays a role in facilitating KDM1A DNA binding and demethylation in HNF4A target genes. However, the authors did not provide a mechanistic explanation for how ZMYM3 mediates the binding of KDM1A to liver-specific TEs. Therefore, the authors should provide more experimental evidence to support this mechanism.

Response:

Thank you for your constructive feedback on our manuscript. In response to your suggestion, we have studied the mechanistic aspect of ZMYM3-mediated facilitation of KDM1A DNA binding, specifically within liver-TEs. Based on our prior findings that ZMYM3 plays a role in mediating the interaction between KDM1A and liver-TEs, we have formulated a hypothesis that liver-TEs may harbor or be proximal to specific DNA sequences recognized by the ZMYM3 protein. To test this hypothesis, we employed the MEME-ChIP program within the MEME suite to identify ZMYM3 motifs and mapped

their locations across the human genome. Subsequently, we calculated the distances between liver-TEs and the identified ZMYM3 motifs by using bedtools software. Our analysis demonstrated a notable overlap or close proximity between most ZMYM3 motifs and liver-TEs (**Fig. S6f**). Importantly, the distances between liver-TEs and ZMYM3 motifs were statistically closer than those observed between a randomly shuffled TE set and the motifs (Wilcox test, $P < 2.2e-16$).

Furthermore, we have provided additional evidence by comparing the KDM1A binding capacity in regions with or without ZMYM3 motifs inside or nearby KDM1A peaks. Our observations revealed a significantly heightened binding capacity of KDM1A in regions harboring ZMYM3 motifs (**Fig. S6G**). This finding suggests a potential role of ZMYM3 in enhancing the DNA binding capability of KDM1A.

9. The reviewer raised the question of whether there is any overlap between liver-specific TE-associated genes and HNF4A target genes. Investigating this overlap would help to elucidate the role of liver-specific TEs in HCC and whether they contribute to the suppression of HNF4A target genes.

Response:

In response to your insightful inquiry, we have incorporated a Venn plot (**Fig. S5c**) to visually represent the intersection between liver-TE-associated genes and HNF4A target genes.

Reviewers' Comments:

Reviewer #1:

Remarks to the Author:

Authors did a great job in addressing all of my concerns. The manuscript is significantly improved and worthy for publication in Nature Communications.

Reviewer #2:

Remarks to the Author:

In the revision of their manuscript "Transposable Elements-Mediated Recruitment of KDM1A Epigenetically Silences HNF4A Expression to Promote Hepatocellular Carcinoma" Jing et al. provide a wealth of additional experimental datasets and analysis to address the reviewer comments. Unfortunately, after reviewing this added information and trying to understand their model for how liver TEs contribute to hepatocellular carcinoma (HCC), there are several points that remain unclear to me:

(1) I am confused about how the "Identification of Liver TEs" analysis was done. Specifically, the ATAC-seq analysis in cancer versus normal tissue and cell lines in Figure 1 b and c, and Supplementary Figure 1 a and b seems over simplified. For example, the number of rows appear to be different in Figure 1 b and c, indicating the TEs that are enriched in normal liver are potentially a subset or partially distinct group of TEs from the 3,762 TEs that were initially identified as "hyper-accessible" in liver cancer. Then in Supplementary Figure 1b (left panel) again the number of rows seems different than in Figure 1 b and c. The authors must account for these differences because as written it sounds like all three analyses included all 3,762 TEs. Also, in the methods it is stated that "We defined transcriptional regulatory regions (TRR) as genomic regions 10kb upstream and downstream of TSS with a Smith-Waterman score >100. We then restricted TEs for analysis to major TE class/families, including SINE (Alu, MIR), LINE (L1, L2), LTR (ERV1, ERVL, ERVL-MaLR), and DNA (hAT-Charlie, TcMar-Mariner) that overlapped with TRR (TRR-TEs)." Which suggests the full list of TEs they used are TRR-TEs, yet in the legend of Supplementary Figure 1b they show the subset of these TEs that (1) overlapped with the TRR and (2) were enriched in the liver Liver-TEs in the left panel, and then define "the transcriptional regulatory regions harboring liver-TEs as the "liver-TE-TRR"" and these are shown in Supplementary Figure 1a and b (right panel). Based on this logic, I can imagine that a single TRR (10 kb upstream and downstream of the TSS) could potentially have multiple TEs, but do not see how a single TE could have multiple TRRs. Yet, there are clearly many more rows in right panel of Supplementary Figure 1 b than the left panel and so I am not sure what is going on here.

(2) In response to my previous concern: "In Figure 1F-I the authors show that removing the TEs near HNF4A increases the expression of HNF4A and that driving hyperactivation of these TEs through recruitment of the dCas9-VP64 fusion protein reduces the expression of HNF4A. However, when the authors increase the activity of these TEs by reducing the activity of KDM1A they see an increase in HNF4A expression. This is logically inconsistent. How can increased activity of presumably the same TEs in one case repress the neighboring gene and in the other case activate the neighboring gene?" the authors state that increased accessibility or activity (induced by CRISPRa) of these elements results in enhanced recruitment of KDM1A (as shown by CUT&Tag) and silencing of HNF4A, and so the data is consistent with the finding that knock down of KDM1A causes an increase in HNF4A. So then does CRISPRa of these TEs cause a decrease in H3K27ac and H3K4me1 of the TEs that are targeted? The finding that you see an increase in accessibility that might be concomitant with a reduction in H3K27ac and H3K4me1 and a reduction in expression of the nearby gene is surprising to me. If this is true, why in Supplementary Figure 1a and b do changes in the accessibility of the TEs mirror changes in accessibility of the nearby regulatory elements? Are the two TEs near HNF4A an exception where increased accessibility causes reduced activity of the nearby promoter? Or normally KDM1A modulates both the TE and the nearby promoter in the same direction and the exception to this rule is due to

ectopic recruitment of CRIPSRa to the TEs in the HNF4A locus which for some reason in this context recruits KDM1A?

Reviewer #3:

Remarks to the Author:

The authors responded the comments from all the reviewers and provided additional experiments. My concerns have been fully addressed.

Reviewer #4:

Remarks to the Author:

While the additional experimental data and explanations provided in this revised manuscript do not fully address all of my concerns, I appreciate the efforts the authors have put into revising their work. I believe the current manuscript is suitable for publication. I have no further comments.

Reviewer #2 (Remarks to the Author):

In the revision of their manuscript “Transposable Elements-Mediated Recruitment of KDM1A Epigenetically Silences HNF4A Expression to Promote Hepatocellular Carcinoma” Jing et al. provide a wealth of additional experimental datasets and analysis to address the reviewer comments. Unfortunately, after reviewing this added information and trying to understand their model for how liver TEs contribute to hepatocellular carcinoma (HCC), there are several points that remain unclear to me:

(1) I am confused about how the “Identification of Liver TEs” analysis was done. Specifically, the ATAC-seq analysis in cancer versus normal tissue and cell lines in Figure 1 b and c, and Supplementary Figure 1 a and b seems over simplified. For example, the number of rows appear to be different in Figure 1 b and c, indicating the TEs that are enriched in normal liver are potentially a subset or partially distinct group of TEs from the 3,762 TEs that were initially identified as “hyper-accessible” in liver cancer. Then in Supplementary Figure 1b (left panel) again the number of rows seems different than in Figure 1 b and c. The authors must account for these differences because as written it sounds like all three analyses included all 3,762 TEs. Also, in the methods it is stated that “We defined transcriptional regulatory regions (TRR) as genomic regions 10kb upstream and downstream of TSS with a Smith-Waterman score >100. We then restricted TEs for analysis to major TE class/families, including SINE (Alu, MIR), LINE (L1, L2), LTR (ERV1, ERVL, ERVL-MaLR), and DNA(hAT-Charlie, TcMar-Mariner) that overlapped with TRR (TRR-TEs).” Which suggests the full list of TEs they used are TRR-TEs, yet in the legend of Supplementary Figure 1b they show the subset of these TEs that (1) overlapped with the TRR and (2) were enriched in the liver Liver-TEs in the left panel, and then define “the transcriptional regulatory regions harboring liver-TEs as the “liver-TE-TRR”” and these are shown in Supplementary Figure 1a and b (right panel). Based on this logic, I can imagine that a single TRR (10 kb upstream

and downstream of the TSS) could potentially have multiple TEs, but do not see how a single TE could have multiple TRRs. Yet, there are clearly many more rows in right panel of Supplementary Figure 1 b than the left panel and so I am not sure what is going on here.

Response

We appreciate your thorough review and are committed to maintaining the highest standards of rigor in our study.

Regarding your concerns about the analyses presented in Figure 1b-c and Supplementary Figure 1b (left panel), we confirm that the entire list of 3762 TEs initially identified as liver-TEs (Supplementary Table 1, the first sheet) is indeed included in these heatmaps. Each heatmap comprises 3762 rows, representing individual TEs within the liver-TE set. To enhance clarity, we have added row annotations alongside the heatmaps to indicate the TE family of each TE. Upon reviewing the revised figures, it is evident that the proportions of TE families are consistent across all three figures.

Moreover, we have provided transparency by including the raw data for the heatmaps in Excel format, available in the compressed source data files. By accessing these data (Fig 1b, Fig 1c, and SFig 1b-left panel), you will find comprehensive information confirming the inclusion of all 3762 TEs in the presented heatmaps, with each row corresponding to a single TE within the liver-TE sets.

In response to your concerns about liver-TE-TRRs and Supplementary Figure 1b (right panel), we apologize for any confusion arising from duplicated or overlapped regions during the generation of the bed file containing liver-TE-TRRs using bedtools software. The presence of duplicated TRR regions may be due to different transcripts sharing the same TSS, while overlapped TRR regions may result from the distance between two TSS being less than 10Kb. We have rectified this issue by merging duplicating and overlapping TRR regions using the "bedtools merge" function, resulting in a refined list comprising 1426 non-overlapped liver-TE-TRR regions, now detailed in Supplementary Table 1 (second sheet). Subsequently, we regenerated the heatmap

presented in Supplementary Figure 1b (right panel) using this corrected list of liver-TE-TRRs. The revised heatmap now consist of 1426 rows, each corresponding to a single independent liver-TE-TRR.

To ensure transparency and facilitate scrutiny, we also included the raw data for Supplementary Figure 1b as source data. Upon examination of the raw data, you will observe that each row in the heatmap corresponds to one of the 1426 liver-TE-TRR regions.

(2) In response to my previous concern: “In Figure 1F-I the authors show that removing the TEs near HNF4A increases the expression of HNF4A and that driving hyperactivation of these TEs through recruitment of the dCas9-VP64 fusion protein reduces the expression of HNF4A. However, when the authors increase the activity of these TEs by reducing the activity of KDM1A they see an increase in HNF4A expression. This is logically inconsistent. How can increased activity of presumably the same TEs in one case repress the neighboring gene and in the other case activate the neighboring gene?” the authors state that increased accessibility or activity (induced by CRISPRa) of these elements results in enhanced recruitment of KDM1A (as shown by CUT&Tag) and silencing of HNF4A, and so the data is consistent with the finding that knock down of KDM1A causes an increase in HNF4A. So then does CRISPRa of these TEs cause a decrease in H3K27ac and H3K4me1 of the TEs that are targeted? The finding that you see an increase in accessibility that might be concomitant with a reduction in H3K27ac and H3K4me1 and a reduction in expression of the nearby gene is surprising to me. If this is true, why in Supplementary Figure 1a and b do changes in the accessibility of the TEs mirror changes in accessibility of the nearby regulatory elements? Are the two TEs near HNF4A an exception where increased accessibility causes reduced activity of the nearby promoter? Or normally KDM1A modulates both the TE and the nearby promoter in the same direction and the exception to this rule is due to ectopic recruitment of CRISPRa to the TEs in the HNF4A locus which for some reason in this context recruits KDM1A?

Response:

Thank you for your insightful comments regarding the effects of manipulating TEs near the HNF4A gene through CRISPRa. We have carefully considered your concerns and conducted further experiments to better understand the underlying mechanisms.

Theoretically, the overall output of chromatin remodeling at this site is determined by the balance between CRISPR activation and the recruited KDM1A-mediated inhibition. In addition to our previous detection of enhanced KDM1A binding (Fig. S3g), as you suggested, we utilized ATAC-seq and CUT&Tag-seq to directly assess changes in chromatin accessibility and histone modifications following CRISPRa. The results clearly revealed that instead of enhancing accessibility, CRISPRa led to a decrease in accessibility, along with reductions in H3K27ac and H3K4me1 modifications in both the promoter and nearby regulatory regions, in the same direction (Fig. S3i). This finding suggests a suppressive effect rather than activation ultimately exhibited by CRISPRa toward the TE site.

We hypothesized that this result might be attributed to the following way: The transient exposure of TEs, induced by CRISPRa, leads to the recruitment of KDM1A. However, the ultimate outcome of CRISPRa was outweighed by the inhibitory effects of recruited KDM1A. To test this hypothesis, we conducted experiments where we knocked down KDM1A and subsequently re-evaluated chromatin accessibility and histone modifications. Indeed, we found that KDM1A knockdown not only restored the decrease in accessibility induced by CRISPRa but also reversed the reduction in H3K27ac and H3K4me1 modifications (Fig. S3i). This observation supports your insightful notion that the ectopic recruitment of CRISPRa to the TEs in the HNF4A locus recruits KDM1A, thus linking the effects of CRISPRa on chromatin state and gene expression to the recruitment of KDM1A.

It is noteworthy that similar scenarios have been reported by other research groups, where targeting regulatory elements with CRISPRa also led to decreased expression (Front Genome Ed. 2023 Oct 25;5:1269115; Mol Cell. 2023 Apr 6;83(7):1125-1139.e8).

The observations collectively reinforce the increasingly recognized feature of CRISPRa-based assays, their effects are heavily reliant on chromatin states and context (Mol Cell. 2023 Apr 6;83(7):1125-1139.e8).

In summary, we propose that when CRISPRa targeted exposes an inhibitory factor attracting element, early in its management, it may inadvertently recruit robust inhibitory factors, such as KDM1A, leading to a suppressive chromatin state. In other words, despite the action initiated by CRISPRa, its impact is outweighed by the subsequent recruitment of KDM1A. This model may explain the observed phenomenon. However, we realized that this phenomenon is distinct from the classical action of CRISPRa. Also, we acknowledge the potential confusion caused by the CRISPRa experiment. In contrast, CRISPR/Cas9-mediated TE depletion might better and robustly elucidate the negative regulatory role of HNF4A-liver-TEs as cis-regulatory elements on HNF4A expression (Fig. 1g and Fig. S1i).

The observed context-dependent effect of CRISPRa in this work relies on KDM1A recruitment, coincidentally emphasizing its role. Therefore, in the revised manuscript, we solely utilized all the CRISPRa experiments as a piece of supplementary evidence to underscore the necessity of KDM1A for chromatin remodeling (Fig. S3g-j). Based on the above findings, we have substantially re-organized this part of results (Page 10, Line 263-277). We have incorporated a dedicated discussion about this issue in the discussion section (Page 16, Line 449-460).

We sincerely appreciate your insightful questions and suggestions, which have prompted us to investigate deeper into the mechanisms underlying the observations. Your input has been invaluable in improving our work.

Reviewers' Comments:

Reviewer #2:

Remarks to the Author:

Through this second round of revision the authors have addressed all of my concerns. I appreciate their added detail in the figure labels, and also the additional experiments to nail down what is happening in response to CRISPRa targeting the TEs upstream of HNF4a. Studying the function of repetitive elements using genomic approaches is inherently challenging, and the level of rigor taken by the authors to investigate the role of TEs in hepatocellular carcinoma make this manuscript truly exceptional. Great work!